# Explaining Grokking and Information Bottleneck through Neural Collapse Emergence

**Keitaro Sakamoto & Issei Sato**
Department of Computer Science
The University of Tokyo
Tokyo, Japan
`{sakakei-1999,sato}@g.ecc.u-tokyo.ac.jp`

## Abstract

The training dynamics of deep neural networks often defy expectations, even as these models form the foundation of modern machine learning. Two prominent examples are grokking, where test performance improves abruptly long after the training loss has plateaued, and the information bottleneck principle, where models progressively discard input information irrelevant to the prediction task as training proceeds. However, the mechanisms underlying these phenomena and their relations remain poorly understood. In this work, we present a unified explanation of such late-phase phenomena through the lens of neural collapse, which characterizes the geometry of learned representations. We show that the contraction of population within-class variance is a key factor underlying both grokking and information bottleneck, and relate this measure to the neural collapse measure defined on the training set. By analyzing the dynamics of neural collapse, we show that distinct time scales between fitting the training set and the progression of neural collapse account for the behavior of the late-phase phenomena. Finally, we validate our theoretical findings on multiple datasets and architectures.

## 1 Introduction

Deep neural networks (DNNs) have demonstrated remarkable success across a variety of tasks, including computer vision, natural language processing, and reinforcement learning, yet their training dynamics under gradient descent often reveal unexpected behavior. In particular, when training continues beyond the point where the training loss has been sufficiently reduced, several intriguing late-phase phenomena have been reported. One such phenomenon is *grokking* (Power et al., 2022), where models initially converge to an overfitting solution that perfectly fits the training data but fails to generalize to unseen data. However, when training continues for a sufficiently long time, the models unexpectedly generalize. Another example is the *information bottleneck* (IB) framework (Tishby et al., 2000; Tishby & Zaslavsky, 2015; Shwartz-Ziv & Tishby, 2017), an information-theoretic perspective on representation learning in DNNs that formalizes the goal of retaining task-relevant information while compressing the task-irrelevant input information. One particularly intriguing observation here is that DNNs do not move directly toward an IB-optimal solution; instead, they first enter a fitting phase where they memorize the training data, followed by a later compression phase in the later training stage during which task-irrelevant input information is discarded.

Taken together, these phenomena share the characteristic that the network evolves toward a more desirable state in the late phase of training, suggesting that some internal change occurs within the training dynamics during this transition. However, the mechanisms driving these phenomena, as well as their relationships, remain poorly understood. Bridging this gap is essential for deepening our understanding of DNN training and for developing more effective training strategies.

In this work, we focus on the geometric structure of the network's representation space and, for the first time, demonstrate that the dynamics of *neural collapse* (Papyan et al., 2020) provide a unified explanation for these late-phase phenomena, offering new insights into their underlying mechanisms. More specifically, the contributions of this work are summarized as follows.

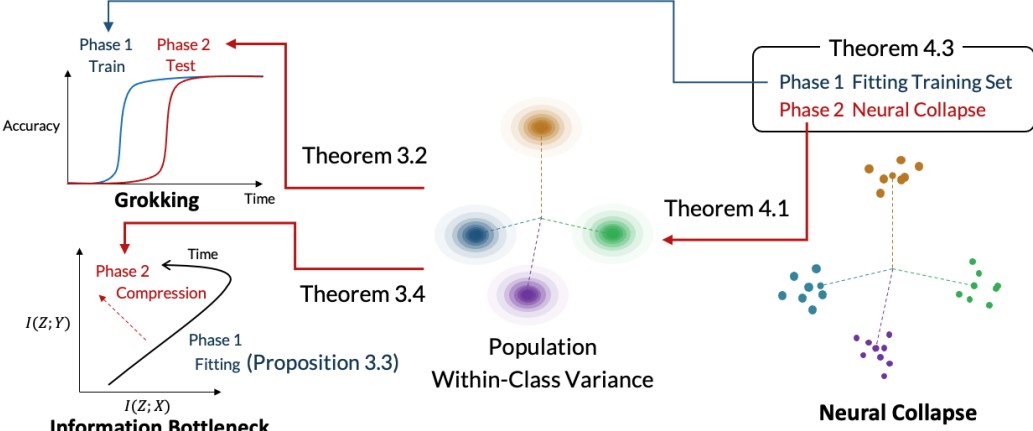

Figure 1: Conceptual relationships in the late-phase training discussed in this work.

- First, we show that the contraction of within-class variance in representations plays a crucial role in both grokking and IB dynamics. Specifically, for grokking, we derive an upper bound on the generalization error in terms of the population within-class variance of the learned representations (Theorem 3.2). Similarly, for the IB principle, we show that the redundant information in the representations, which is discarded in the IB compression phase, is bounded by the population within-class variance (Theorem 3.4). These results motivate the analysis of neural collapse, whose properties include the collapse of empirical within-class representations in the training data, known as NC1 (Section 3).

- Second, we provide a quantitative analysis of the discrepancy between the population within-class variance and its empirical counterpart, using an approach analogous to generalization error analysis (Theorem 4.1). This allows us to evaluate to what extent the reduction of empirical within-class variance, that is, the progression of neural collapse, implies a corresponding reduction in the population within-class variance, a key quantity we identified. In this way, we relate the behaviors of grokking and IB dynamics to the development of neural collapse (Section 4.1).

- Finally, building on the preceding results, we analyze the development of neural collapse by explicitly tracking the dynamics of gradient descent. Leveraging the results of Jacot et al. (2025), we establish that the empirical within-class variance decreases during training and characterize its time scale (Theorem 4.3). In particular, depending on the strength of weight decay, the time scale on which neural collapse is sufficiently realized can lag far behind the time scale of fitting the training data. This result suggests that the timing of neural collapse emergence underlies the delayed generalization in grokking as well as the compression phase in IB dynamics (Section 4.2).

These relationships are illustrated in Figure 1, along with the corresponding theorems and propositions presented in the main text. In addition to the theoretical analysis, we validate our findings through extensive experiments on various datasets and architectures. Taken together, our work deepens the understanding of late-phase training phenomena from the perspective of neural collapse. [1]

**Notation.** We consider a $K$-class classification problem with a dataset $S = \{(\boldsymbol{x}_i, y_i)\}_{i=1}^N$ consisting of $N$ examples, where $\boldsymbol{x}_i \in \mathbb{R}^d$ is the input and $y_i \in [K](:= \{1, 2, \ldots, K\})$ is the class label. The input domain is $\mathcal{X} \subseteq \mathbb{R}^d$, and $\boldsymbol{y}_i$ denotes the one-hot encoding label in $\{0, 1\}^K$. Let $\boldsymbol{X} \in \mathbb{R}^{d \times N}$ and $\boldsymbol{Y} \in \{0, 1\}^{K \times N}$ denote the training inputs and one-hot labels, respectively. We use $S_c = \{(\boldsymbol{x}_i, y_i) \in S \mid y_i = c\}$ to denote the subset of class-$c$ examples, and, by abuse of notation, also the corresponding index set $\{i \in [N] \mid y_i = c\}$. We denote its cardinality as $n_c = |S_c|$. Let $X$ and $Y$ denote the random variables representing the input and label, respectively. We write $p_{X,Y}$ for the joint distribution over $(X, Y)$, and $p_X$ and $p_Y$ for their marginal distributions. When the variables are clear from the context, we simply write $p$ to refer to the corresponding distribution. Let $I(X; Y)$ be the mutual information (MI) between random variables $X$ and $Y$. We denote the multivariate Gaussian distribution with mean $\boldsymbol{\mu}$ and covariance matrix $\boldsymbol{\Sigma}$ as $N(\boldsymbol{\mu}, \boldsymbol{\Sigma})$. We use $\log(\cdot)$

---

[1]Code is available at https://github.com/keitaroskmt/collapse-dynamics.

to denote the natural logarithm. For linear-algebraic notation, for a matrix $\boldsymbol{A}$, we use $\|\boldsymbol{A}\|_2$ to denote the spectral norm, and $\|\boldsymbol{A}\|_{2,1}$ to denote the $(2,1)$ matrix norm, defined as $\|\boldsymbol{A}\|_{2,1} = \sum_i \|\boldsymbol{a}_i\|_2$, where $\boldsymbol{a}_i$ is the $i$-th column of $\boldsymbol{A}$. Finally, we use $\langle \boldsymbol{a}, \boldsymbol{b} \rangle = \boldsymbol{a}^\top \boldsymbol{b}$ to denote the standard Euclidean inner product.

## 2 RELATED WORK

**Grokking.** On the empirical side, several studies attempted to explain the cause of grokking in terms of the parameter compression (Liu et al., 2023a; Varma et al., 2023) and some complexity measures (Nanda et al., 2023; Liu et al., 2023b; Humayun et al., 2024; DeMoss et al., 2025). Other studies sought to relate grokking to other training-related concepts, such as double descent (Davies et al., 2023; Huang et al., 2024) and optimization stability (Thilak et al., 2022). On the theoretical side, a high-dimensional limit of the linear model was analyzed in the setting of regression (Levi et al., 2024) and binary classification (Beck et al., 2025). There are also studies analyzing two-layer networks with XOR data (Xu et al., 2024) and mean-field analysis (Rubin et al., 2024). The most closely related line of work focuses on the transition from the kernel regime to the rich regime (Lyu et al., 2024; Kumar et al., 2024). Our study provides a novel perspective based on the emergence of neural collapse, offering a new connection to the representation-learning view of Liu et al. (2022b).

**Information Bottleneck.** The IB principle (Tishby et al., 2000) has been analyzed in the context of deep learning by modeling the successive neural network layers as a Markov chain (Tishby & Zaslavsky, 2015; Shwartz-Ziv & Tishby, 2017; Saxe et al., 2018; Geiger, 2021; Lorenzen et al., 2022; Adilova et al., 2023; Butakov et al., 2024a;b). In addition to these studies analyzing DNN training from the IB perspective, minimizing the IB objective has also been shown to benefit generalization bounds (Kawaguchi et al., 2023; Sefidgaran et al., 2023). Several studies have attempted to explain the IB dynamics. For example, Shwartz-Ziv et al. (2019) attributed the compression phase to the diffusion component of stochastic gradient descent (SGD), while Goldfeld et al. (2019) and Koch & Ghosh (2025) discussed its relation to geometric compression and grokking, respectively. However, these studies lack the rigorous theoretical analysis of the IB dynamics. A related line of work analyzes late-phase behavior through reconstruction loss rather than mutual information (Schneider & Prabhushankar, 2024; Schneider, 2025), though this perspective provides only an indirect proxy for the IB dynamics. In contrast, our work offers a new explanation of IB dynamics based on neural collapse, a geometric form of compression.

**Neural Collapse.** Neural collapse (Papyan et al., 2020) is a late-stage training phenomenon where the representations of the training data converge to a simplex equiangular tight frame (ETF) formed with the class mean representations. A major line of research analyzes the optimality of such solutions under an unconstrained feature model (UFM) and its variants (Fang et al., 2021; Mixon et al., 2022; Lu & Steinerberger, 2022; Tirer & Bruna, 2022; Thrampoulidis et al., 2022; Dang et al., 2023; Tirer et al., 2023; Súkeník et al., 2023; 2024; Jiang et al., 2024), where the features are treated directly as optimization variables. Several works have examined the learning dynamics toward neural collapse under UFM (Zhu et al., 2021; Mixon et al., 2022; Ji et al., 2022; Zhou et al., 2022a;b), but this setting cannot reflect the input data properties and diverges from the actual parameter-based training dynamics. Beyond UFM setting, recent studies instead analyze parameter updates under weight regularization, opening a promising new direction (Jacot et al., 2025; Wu & Mondelli, 2025). Building on this, our study broadens the understanding of training dynamics by revealing the connection between late-phase phenomena and neural collapse.

## 3 EMERGENT BEHAVIOR IN LATE-PHASE TRAINING

In this section, we examine two intriguing phenomena that arise in the late-phase training and remain active topics of research: grokking and IB dynamics. Through separate analysis in the following subsections, we demonstrate that the population within-class variance in the representation space plays a critical role and constitutes a unifying factor. To this end, in the following, let $g : \mathbb{R}^d \to \mathbb{R}^{d_{\text{rep}}}$ denote the feature extractor, which may take various model architectures, $\boldsymbol{W} \in \mathbb{R}^{K \times d_{\text{rep}}}$ the last layer classifier, and $f : \mathbb{R}^d \to \mathbb{R}^K$ with $f(\boldsymbol{x}) = \boldsymbol{W} g(\boldsymbol{x})$ the full model. We use $f(\boldsymbol{x})_i$ to denote the $i$-th component of the prediction vector $f(\boldsymbol{x})$.

We now introduce the population within-class variance, which will play a central role in this section. We remark that this metric is not introduced as an ad-hoc measure; rather, it naturally arises from our analysis of the second-phase dynamics presented below. In particular, the scale-invariant formulation in Definition 3.1 ensures that this quantity reflects genuine geometric concentration, disentangled from changes in output scale. This motivates the following definition.

**Definition 3.1** (Population within-class variance). To evaluate variance in a scale-invariant manner, we consider the expected within-class variance of the rescaled feature extractor, defined as

$$\mathbb{E}_{X|Y=c}\left[\left\|\tilde{g}(X) - \mathbb{E}_{X|Y=c}\left[\tilde{g}(X)\right]\right\|_2^2\right], \text{ where } \tilde{g}(\boldsymbol{x}) = \frac{g(\boldsymbol{x})}{B_g}, \ B_g = \sup_{\boldsymbol{x}\in\mathcal{X}} \|g(\boldsymbol{x})\|_2.$$

This quantity, as well as its expectation over the label distribution $p_Y$, serves as a key component in the results of this section. The results established here using the population within-class variance then motivate the subsequent analysis of neural collapse, which characterizes the reduction of empirical within-class variance in the training data.

## 3.1 GROKKING: PHASE 2 GENERALIZATION DYNAMICS

In the context of grokking, Liu et al. (2022b) empirically shows that the acquisition of structured representations leads to a transition from memorization to generalization, which we further analyze here. As a simple observation, since the network output is given by $f(\boldsymbol{x}) = \boldsymbol{W}g(\boldsymbol{x})$, achieving good generalization is facilitated when the representations of each class are linearly separable in distribution. When the training loss is sufficiently reduced at some time step $\tau_1$, the existence of a weight matrix $\boldsymbol{W}(\tau_1)$ indicates that the representations of the training set can be linearly separable; however, this does not guarantee that the representations in distribution are linearly separable. Intuitively, when the representations of each class are more tightly clustered, achieving linear separability becomes easier, and better generalization performance can be expected. Formalizing this intuition yields the following theorem.

**Theorem 3.2** (Generalization via population within-class variance). *For a fixed feature extractor $g$ and the last layer $\boldsymbol{W} = (\boldsymbol{w}_1, \ldots, \boldsymbol{w}_K)^\top$, we have*

$$\Pr\left(\underset{i\in[K]}{\arg\max}\left\{f(\boldsymbol{x})_i\right\} \neq y\right) \leq \sum_{c=1}^K p_Y(c) \sum_{k\neq c}\left(1 + \frac{\max\left\{\left\langle\mathbb{E}_{X|Y=c}[\tilde{g}(X)], \frac{\boldsymbol{w}_c-\boldsymbol{w}_k}{\|\boldsymbol{w}_c-\boldsymbol{w}_k\|_2}\right\rangle, 0\right\}^2}{\mathbb{E}_{X|Y=c}\left[\left\|\tilde{g}(X) - \mathbb{E}_{X|Y=c}[\tilde{g}(X)]\right\|_2^2\right]}\right)^{-1}.$$

This theorem follows directly from applying the union bound and a variance-based tail probability bound. The proof is given in Appendix B.1. Theorem 3.2 states that improving test accuracy is facilitated by two conditions: i) the class mean representations $\mathbb{E}_{X|Y=c}[\tilde{g}(X)]$ become more aligned with the corresponding last-layer weights $\boldsymbol{w}_c$, and ii) the population within-class variance decreases. From a grokking perspective, when the training loss is sufficiently reduced, it remains unclear how well the classifier $\boldsymbol{W}$ aligns with the class means of the representations. Nevertheless, independent of the properties of $\boldsymbol{W}$, reducing the within-class variance of the representations tightens the upper bound on the generalization error. To uncover the mechanism of grokking, it remains to show that the within-class variance decreases even after the training accuracy has been sufficiently improved.

## 3.2 IB DYNAMICS

The IB principle (Tishby et al., 2000) formulates a constrained optimization problem: it seeks a compact representation $Z$ of the input $X$ that retains as much information as possible about the target $Y$ while compressing $X$. This formulation is built on the idea that a concise short code can extract the features of $X$ essential for predicting $Y$, and the IB principle serves as one approach to explaining the representations acquired by DNNs. Under the Markov chain $Y \to X \to Z$, this can be formulated using MI as finding the conditional distribution $p(Z|X)$ that minimizes

$$\min_{p(Z|X)} I(Z;X) - \beta I(Z;Y), \tag{1}$$

where $\beta > 0$ is a trade-off parameter controlling the balance between compression and information preservation. When analyzing DNNs within the IB framework, we encounter several difficulties:

for a deterministic network and a continuous representation, $I(Z;X)$ can be infinite, making the analysis ill-defined; furthermore, the network parameters are not reflected in the IB analysis (Saxe et al., 2018; Amjad & Geiger, 2019; Goldfeld et al., 2019). To address these issues, we analyze the representation after independently adding an arbitrarily small Gaussian noise $E \sim N(\mathbf{0}, \sigma^2 \mathbf{I})$, $\sigma \ll 1$, which is a standard approach in IB analysis of DNNs (Saxe et al., 2018; Goldfeld et al., 2019; Butakov et al., 2024b). It should be noted that such a small amount of noise has a negligible effect on the network's output, making it a good proxy for the actual network. Since we are currently focusing on the output of the feature extractor $g$, we denote the representation as $Z = g(X) + B_g E$, where $B_g = \sup_{\boldsymbol{x} \in \mathcal{X}} \|g(\boldsymbol{x})\|_2$ denotes the output scale of $g$ introduced in Definition 3.1. We use $\boldsymbol{z}$ to denote the realization of $Z$.

Representation dynamics in the two-dimensional $(I(Z;X), I(Z;Y))$ plane, which is called *information plane*, is a useful tool for analyzing the training dynamics of DNNs from the IB perspective (Shwartz-Ziv & Tishby, 2017). As a preliminary observation, to perform classification accurately with a neural network, both $I(Z;X)$ and $I(Z;Y)$ need to be sufficiently large; this follows from the following proposition (see Appendix B.2 for the proof):

**Proposition 3.3** (Phase 1 of IB dynamics). *For any last-layer classifier $\boldsymbol{W}$, let $\ell_{CE}(y, \boldsymbol{z})$ denote the cross-entropy loss between the target $y \in [K]$ and the predicted logits $\boldsymbol{W}\boldsymbol{z} \in \mathbb{R}^K$, defined by $\ell_{CE}(y, \boldsymbol{z}) = -\log\left(\exp((\boldsymbol{W}\boldsymbol{z})_y)/\sum_{c=1}^{K}\exp((\boldsymbol{W}\boldsymbol{z})_c)\right)$. Then, we have*

$$I(Z;X) \geq I(Z;Y) \geq -\mathbb{E}_{Y,Z}\left[\ell_{CE}(Y, Z)\right] + const.$$

Proposition 3.3 implies that if the network at initialization discards information about both $X$ and $Y$, the training process must appropriately increase $I(Z;Y)$, and consequently $I(Z;X)$, over time. Please note that if the initial state of the network sufficiently preserves information about $X$ and $Y$ without collapsing the outputs, then this MI increase phase is unnecessary.

An intriguing observation in the existing information plane work is that, in the late stage of training, DNNs tend to compress $I(Z;X)$ while preserving $I(Z;Y)$, thereby moving toward a more optimal solution with respect to the IB objective in Equation (1). We explain this behavior through the following theorem, which is based on the degree of collapse of the population within-class variance, a quantity introduced in Definition 3.1.

**Theorem 3.4** (Phase 2 of IB dynamics via population within-class variance). *Let $Z = g(X) + B_g E$, where $E \sim N(\mathbf{0}, \sigma^2 \mathbf{I})$. Here, the variance $\sigma^2 > 0$ is chosen to be small enough to ensure a negligible effect on the network output. Then, the superfluous information is bounded as follows:*

$$I(Z;X) - I(Z;Y) = I(Z;X|Y) \leq \frac{1}{2\sigma^2}\mathbb{E}_{X,Y}\left[\left\|\tilde{g}(X) - \mathbb{E}_{X|Y}\left[\tilde{g}(X)\right]\right\|_2^2\right].$$

The proof is given in Appendix B.2. We further show the tightness and behavior of this upper bound in Proposition B.3. Theorem 3.4 highlights the role of within-class variance in reducing superfluous information, corresponding to the evolution toward the upper-left in the information plane. Together with Theorem 3.2, the analysis in this section establishes population within-class variance as a key measure. In the next section, we examine how this measure decreases during training and, for grokking, how it proceeds on a different time scale from fitting training set.

# 4 EVOLUTION OF WITHIN-CLASS VARIANCE

In this section, motivated by the previous results, we analyze the training dynamics of within-class variance in the learned representation space. We first reduce the discussion of within-class variance to neural collapse, namely the geometric arrangement of class-wise representations in the training set. We then examine how neural collapse progresses under gradient descent, independently of the training loss convergence. These results together shed light on the evolution of within-class variance.

## 4.1 POPULATION WITHIN-CLASS VARIANCE AND NEURAL COLLAPSE

Building on Theorems 3.2 and 3.4, we examine how the upper bounds based on within-class variance can be approximated using training data. Since this requires delving into the specifics of the trained

model, for the remainder of the paper we consider the following standard DNN:

$$f(\boldsymbol{x}) = \boldsymbol{W}_L g(\boldsymbol{x}) = \boldsymbol{W}_L \sigma_{\text{out}} \left( \boldsymbol{W}_{L-1} \sigma \left( \cdots \sigma \left( \boldsymbol{W}_1 \boldsymbol{x} \right) \cdots \right) \right), \tag{2}$$

where $\boldsymbol{W}_\ell \in \mathbb{R}^{d_\ell \times d_{\ell-1}}$, with $d_0 = d$ denoting the input dimension. Note that $\boldsymbol{W}_L$ and $d_{L-1}$ respectively correspond to $\boldsymbol{W}$ and $d_{\text{rep}}$ used in the previous section. Here $\sigma$ denotes the element-wise activation function with $\sigma(0) = 0$ and 1-Lipschitz, while $\sigma_{\text{out}}$ is the activation at the output of the feature extractor $g$. Throughout Section 4.1, we consider the standard setting $\sigma_{\text{out}} = \sigma$, and in Section 4.2, we set $\sigma_{\text{out}} = \text{id}$, i.e., the identity map, for analytical convenience.

We first provide a formal bound on the difference between the population within-class variance, which served as an important measure in the previous section, and its empirical counterpart. The following result is based on a standard approach of uniform convergence commonly used in generalization error analysis. The proof is provided in Appendix B.3.

**Theorem 4.1** (Concentration of within-class variance). *Suppose the input domain $\mathcal{X}$ is bounded, i.e., $\|\boldsymbol{x}\|_2 \leq B_x$ for all $\boldsymbol{x} \in \mathcal{X}$. Let $\delta \in (0,1)$ and $d_{\max} = \max_{0 \leq \ell \leq L-1} d_\ell$, and recall $B_g = \sup_{\boldsymbol{x} \in \mathcal{X}} \|g(\boldsymbol{x})\|_2$. Define the complexity measures of $g$ as $\Pi(g) = \max\left\{ \prod_{\ell=1}^{L-1} \|\boldsymbol{W}_\ell\|_2, 1 \right\}$ and $\Lambda(g) = \max\left\{ \left( \sum_{\ell=1}^{L-1} \left( \|\boldsymbol{W}_\ell\|_{2,1} / \|\boldsymbol{W}_\ell\|_2 \right)^{2/3} \right)^{3/2}, 1 \right\}$. Then, with probability at least $1 - \delta$, for all $c \in [K]$, we have*

$$\left| \mathbb{E}_{X|Y=c} \left[ \left\| \tilde{g}(X) - \mathbb{E}_{X|Y=c} \left[ \tilde{g}(X) \right] \right\|_2^2 \right] - \frac{1}{n_c} \sum_{i \in S_c} \left\| \tilde{g}(\boldsymbol{x}_i) - \frac{1}{n_c} \sum_{j \in S_c} \tilde{g}(\boldsymbol{x}_j) \right\|_2^2 \right|$$

$$\leq O\left( \frac{1}{\sqrt{n_c}} \left[ \frac{1}{\sqrt{n_c}} + \frac{\Pi(g) B_x}{B_g} \log(n_c) \sqrt{\log(d_{\max})} \Lambda(g) + \sqrt{\log(K/\delta) + \log\log\left( \Pi(g)\Lambda(g) \right)} \right] \right).$$

Note that in $\Pi(g)B_x/B_g$, both the numerator and the denominator represent the output scale of $g$; the numerator is a spectral-norm-based upper bound, while the denominator reflects the actual output scale. This theorem establishes an $O(1/\sqrt{n_c})$ bound on the gap between the population within-class variance studied earlier and the empirical within-class variance in the training data. This guarantee justifies focusing on the empirical variance in the subsequent analysis, where we investigate its dynamics as a proxy for the population behavior.

## 4.2 Neural Collapse Dynamics

Analyzing the variance of class-wise representations in the training data naturally motivates the study of neural collapse. We therefore begin by defining the neural collapse metrics. Neural collapse refers to several characteristic properties in the late stage of training: (NC1) the representations collapse to their respective class means; (NC2) the class means form an ETF; and (NC3) each row of $\boldsymbol{W}_L$ aligns with the corresponding class mean up to a positive scaling.

Let $\boldsymbol{\mu}_c = \frac{1}{n_c} \sum_{i \in S_c} g(\boldsymbol{x}_i)$ and $\boldsymbol{\mu}_G = \frac{1}{N} \sum_{i \in S} g(\boldsymbol{x}_i)$, and denote their counterparts for $\tilde{g}$ as $\tilde{\boldsymbol{\mu}}_c$ and $\tilde{\boldsymbol{\mu}}_G$. Among neural collapse metrics, we focus on NC1, defined as $\text{NC1} = \frac{\text{Tr}(\boldsymbol{\Sigma}_W)}{\text{Tr}(\boldsymbol{\Sigma}_B)}$, where the within-class covariance $\boldsymbol{\Sigma}_W = \frac{1}{N} \sum_{c=1}^{K} \sum_{i \in S_c} \left( g(\boldsymbol{x}_i) - \boldsymbol{\mu}_c \right) \left( g(\boldsymbol{x}_i) - \boldsymbol{\mu}_c \right)^\top$ and the between-class covariance $\boldsymbol{\Sigma}_B = \frac{1}{K} \sum_{c=1}^{K} \left( \boldsymbol{\mu}_c - \boldsymbol{\mu}_G \right) \left( \boldsymbol{\mu}_c - \boldsymbol{\mu}_G \right)^\top$. Since we consider the within-class variance rescaled instead of dividing by the between-class variance, as appears in Theorem 4.1, we define the following measure as the rescaled NC1 (RNC1):

$$\text{RNC1} := \frac{1}{B_g^2} \text{Tr}\left( \boldsymbol{\Sigma}_W \right) = \frac{1}{N} \sum_{c=1}^{K} \sum_{i \in S_c} \left\| \tilde{g}(\boldsymbol{x}_i) - \tilde{\boldsymbol{\mu}}_c \right\|_2^2.$$

**Remark 4.2** (Difference between RNC1 and NC1). While both metrics are invariant to the scale of $g$, NC1, which is normalized by the between-class variance, does not reduce to zero when all features collapse to a single point. This reflects class-center separation, a property already captured by NC2, whereas RNC1 isolates the within-class variance aspect more directly than NC1.

We next specify the training setup for analyzing the dynamics of neural collapse. The network $f$ is trained by gradient descent with a step size $\eta > 0$. The loss function is the squared loss with weight

decay, controlled by a hyperparameter $\lambda > 0$, and is defined as follows:

$$\boldsymbol{\theta}(\tau + 1) = \boldsymbol{\theta}(\tau) - \eta \nabla_{\boldsymbol{\theta}} \widehat{\mathcal{L}}_\lambda(\boldsymbol{\theta}(\tau)), \quad \widehat{\mathcal{L}}_\lambda(\boldsymbol{\theta}(\tau)) = \frac{1}{2} \|f_\tau(\boldsymbol{X}) - \boldsymbol{Y}\|_F^2 + \frac{\lambda}{2} \sum_{\ell=1}^L \|\boldsymbol{W}_\ell(\tau)\|_F^2,$$

where $\boldsymbol{\theta}$ is the concatenation of all network parameters $\{\boldsymbol{W}_\ell\}_{\ell=1}^L$, and $f_\tau$ is the network defined in Equation (2) at time step $\tau$. Accordingly, we denote the value of RNC1 at time step $\tau$ as $\mathrm{RNC1}(\tau)$.

We now analyze the dynamics of neural collapse measured in terms of RNC1, together with the convergence of the training loss. Our results build on the recent results of Jacot et al. (2025), which assume (i) a pyramidal network architecture, (ii) a smooth activation function, and (iii) a specific initialization condition; their formal statements are given in Appendix B.4. The next theorem establishes that the training loss and RNC1 converge on different time scales. Here we use the standard Big-O notations $O(\cdot)$ and $\Omega(\cdot)$ to describe the dependence on $\lambda$, $\eta$, $\epsilon_1$, and $\epsilon_2$.

**Theorem 4.3** (Time scales of neural collapse dynamics). *Suppose that the network $f$ satisfies Assumptions B.3 to B.5 and that the input domain $\mathcal{X}$ is bounded. Fix $0 < \epsilon_1 < \frac{1}{8} \min_{c \in [K]} n_c$ and $\epsilon_2 > 0$. For weight decay $\lambda = O(\epsilon_1)$, learning rate $\eta = O(\epsilon_2)$, and time steps $\tau_1 < \tau_2$ satisfying*

$$\tau_1 = \Omega\left(\frac{1}{\eta} \log \frac{1}{\epsilon_1}\right), \quad \tau_2 = \Omega\left(\frac{1}{\lambda \eta} \log \frac{1}{\epsilon_2}\right),$$

*the regularized training loss and RNC1 are bounded as*

$$\widehat{\mathcal{L}}_\lambda(\boldsymbol{\theta}(\tau_1)) \leq \epsilon_1, \quad \mathrm{RNC1}(\tau_2) = O(\epsilon_1 + \epsilon_2).$$

This theorem not only shows that RNC1 indeed decreases under gradient descent training, but also clarifies the time scales that govern the convergence of the training loss and RNC1. For the target thresholds $\epsilon_1$ and $\epsilon_2$, both require a logarithmic order of training steps in their reciprocals. On the other hand, while the convergence of the training loss is independent of the weight decay parameter $\lambda$, the time scale for the convergence of RNC1 grows inversely with smaller $\lambda$. This implies that $\tau_2$ can be much larger than $\tau_1$ depending on the value of $\lambda$, indicating that the convergence of RNC1 may occur substantially later than that of the training loss.

**Remark 4.4** (Summary of Theoretical Results). Up to this point, as summarized in Figure 1, we have developed our analysis starting from the late-phase phenomena. We now provide an overall summary of the insights obtained from our theoretical results in Sections 3 and 4.

**Grokking.** By combining Theorems 3.2 and 4.1, we showed that the decrease of the empirical within-class variance, namely RNC1, leads to improved test accuracy. Theorem 4.3 further established the time scale governing this decrease, jointly with the convergence of the training loss. In particular, when $\tau_1 \ll \tau_2$ in Theorem 4.3, for example with a small weight decay $\lambda$, neural collapse occurs later than the convergence of the training loss, and generalization improvement lags behind fitting training set; that is the grokking behavior.

**IB Dynamics.** For the first fitting phase, Proposition 3.3 demonstrated that this phase is necessary whenever the network discards input information. Unlike the first phase of grokking, this fitting phase is not necessarily tied to training loss convergence. For the second compression phase, Theorems 3.4 and 4.1 showed that it proceeds together with the decrease of RNC1, whose convergence and time scale were established in Theorem 4.3.

## 5 EXPERIMENTS

In this section, we conduct experiments to validate the theoretical results in Sections 3 and 4.

### 5.1 GROKKING

We first analyze the relationship between grokking, within-class variance, and neural collapse. Following Liu et al. (2023a), we train an MLP on the MNIST dataset (LeCun et al., 2010). The model has four layers with architecture $[784, 200, 200, 200, 10]$ and ReLU activation. The initialization scale is increased by a factor of $8$ as in Liu et al. (2023a), and we use the AdamW optimizer

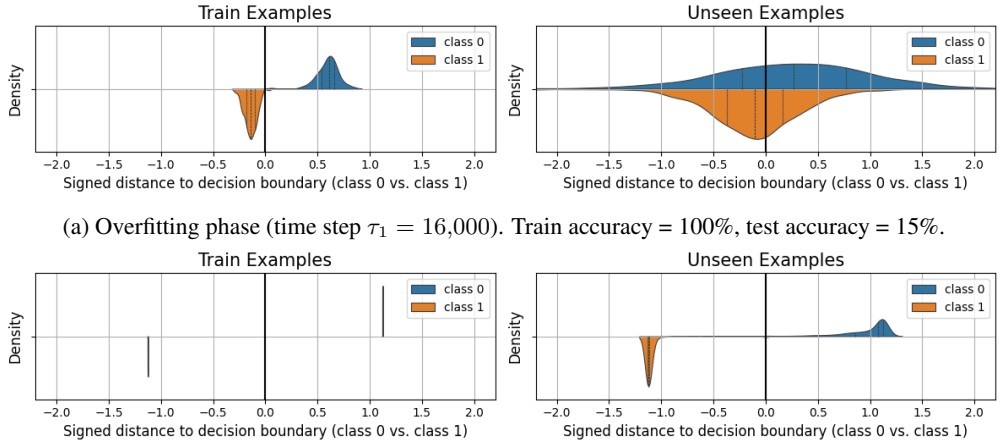

(a) Overfitting phase (time step $\tau_1 = 16{,}000$). Train accuracy = 100%, test accuracy = 15%.

(b) Convergence phase (time step $\tau_2 = 100{,}000$). Train accuracy = 100%, test accuracy = 88%.

Figure 2: Margins of individual examples at two time steps during grokking. The margin of each example is defined as the signed distance from its representation to the decision boundary determined by the last-layer classifier, calculated as $\left(\langle \boldsymbol{w}_0 - \boldsymbol{w}_1, g(\boldsymbol{x})\rangle + b_0 - b_1\right)/\|\boldsymbol{w}_0 - \boldsymbol{w}_1\|_2$, where $b_c$ denotes the bias term for class $c$. We trained a 4-layer MLP on the MNIST dataset. These results reveal the link between representation variance and generalization, supporting Theorems 3.2 and 4.1. We additionally provide similar visualizations for several other class pairs in Appendix D.1.

(Loshchilov & Hutter, 2019) with a learning rate of $1e-3$ and weight decay of $0.01$. Additional results on other datasets and architectures, including convolutional neural networks (CNNs) and Transformers (Vaswani et al., 2017), are presented in Appendix D.1. In examining the dynamics of neural collapse, we primarily track the RNC1 score, which is the focus of our theoretical analysis. As an additional geometric metric, we also report the NC2 score, another geometric aspect of neural collapse. We define it as the condition number of the matrix of class mean vectors, $\mathrm{NC2} = \kappa\left((\boldsymbol{\mu}_1, \ldots, \boldsymbol{\mu}_K)\right)$. We include NC2 to provide supplementary geometric insight into the arrangement of the class mean vectors.

We begin by motivating our approach of analyzing grokking through the lens of representation learning. Figure 2 shows how the representations of training and unseen examples evolve in a setting where grokking occurs. In the overfitting phase (top), the training examples are separated, but the unseen examples exhibit large within-class variance despite their mean shifting toward the correct class, resulting in many misclassifications. As training proceeds, the training examples become further separated and collapse into single points (bottom). At this stage, as discussed in Theorem 4.1, the collapse of the training representations is to some extent inherited by the underlying distribution, leading to the reduction of the population within-class variance, as illustrated in the right panel. Consequently, test accuracy improves, and grokking emerges in a manner consistent with the generalization bound established in Theorem 3.2.

Next, Figure 3 presents the grokking dynamics as well as those of RNC1 and NC2 scores under different weight decays, which further supports our analysis in two major respects. **First**: the decrease of RNC1 is synchronized not with fitting the training set but with the emergence of grokking, i.e., the improvement of test accuracy. This phenomenon consistently appears across all weight-decay settings, reinforcing our theoretical result that explains grokking through the progression of neural collapse. Although NC2 is not directly treated in our analysis, it exhibits almost the same behavior as RNC1 and converges toward 1, indicating the emergence of neural collapse. **Second**: stronger weight decay $\lambda$ accelerates the timing of grokking and narrows the gap between training accuracy saturation and generalization. This observation aligns with our main result linking grokking to RNC1 dynamics and also supports Theorem 4.3, which shows that the convergence of the RNC1 score becomes faster as the weight decay increases.

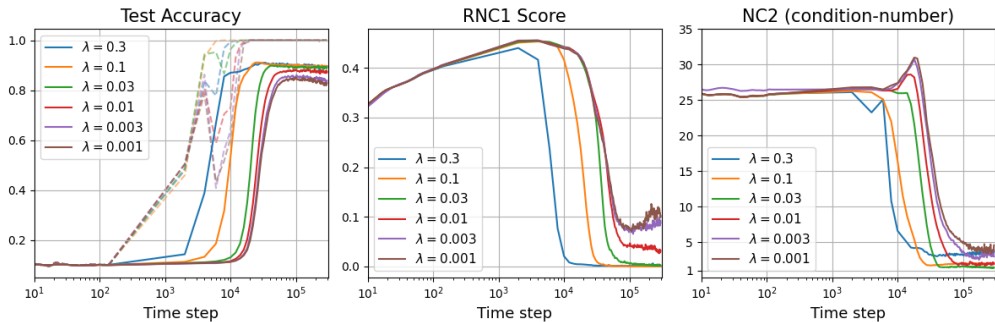

Figure 3: Dynamics of test accuracy, RNC1, and NC2 scores throughout training for different weight decay coefficients $\lambda$. In the test accuracy panel (left), the training accuracy is additionally shown in dashed lines of the same color to visualize grokking behavior. Results are averaged over five different seeds with an MLP trained on the MNIST dataset. These results demonstrate the connection between neural collapse and grokking, and their time scales, supporting Theorems 3.2, 4.1 and 4.3.

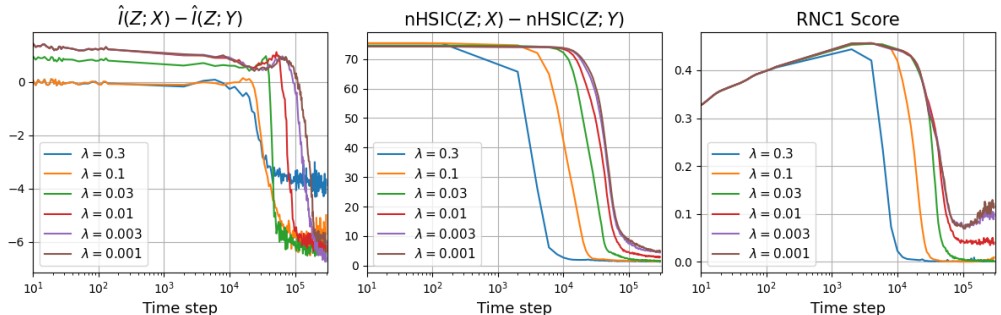

Figure 4: Dynamics of redundant information in IB framework (estimated via MI and nHSIC) and RNC1 scores throughout training for different weight decay $\lambda$. Results are averaged over five different seeds with an MLP trained on the MNIST dataset. These results show the connection between neural collapse and IB dynamics, as well as their time scales, supporting Theorems 3.4, 4.1 and 4.3.

## 5.2 IB DYNAMICS

In this section, we conduct experiments on IB dynamics. Using the same setup as in the previous experiments, we measure how the MI between the learned representation $Z$ and the input $X$, as well as between $Z$ and the target $Y$, evolves during training. A fundamental difficulty in IB experiments with DNNs is that, beyond toy settings, conventional estimators based on binning or kernel density estimation fail to provide accurate estimates in high-dimensional input or representation due to the curse of dimensionality. This challenge remains an active research area, and in this work, we adopt the recent dimensionality-reduction-based MI estimator of Butakov et al. (2024b). Specifically, we first compress the variables into four dimensions and then estimate each entropy term of MI using the k-NN-based Kozachenko-Leonenko method (Kozachenko, 1987; Berrett et al., 2019). We denote this estimate by $\hat{I}$ and provide the details in Appendix C. In addition, to support the reliability of our experimental findings and to address the inherent difficulty of MI estimation, we use the normalized Hilbert-Schmidt independence criterion (nHSIC) (Gretton et al., 2005), a proxy widely adopted in information-theoretic analysis of DNNs. Please see Appendix C for the background.

Figure 4 shows, under the same setting as in Figure 3, the behavior of redundant information in the IB principle discussed in Section 3.2 together with the corresponding RNC1 score. As shown in the grokking experiments, the decrease of the RNC1 score in the later training stage occurs earlier when the weight decay is stronger (right), which is consistent with Theorem 4.3. Theorems 3.4 and 4.1 establish that this decrease in the RNC1 score contributes to the reduction of the redundant information, and the figure demonstrates this result. The left figure shows MI estimates, indicating that the stronger weight decay accelerates the decrease of redundant information. Although the

information reduction is slightly delayed for large weight decay values ($\lambda = 0.3, 0.1$), the decrease of RNC1 scores actually leads to the reduction of redundant information. Since MI is estimated by decomposing it into differential entropy terms that are estimated separately, the resulting MI estimates can take negative values despite the non-negativity of MI, while still capturing the overall decreasing trend. To further support our findings, we also include the results using nHSIC to measure superfluous information (middle). This result exhibits the same qualitative behavior as the RNC1 score and corroborates our theoretical analysis of the IB dynamics. Additional results for other model architectures and datasets are provided in Appendix D.2.

## 6 CONCLUSION

In this work, we focus on grokking and IB dynamics as two representative late-phase phenomena of DNNs, whose mechanisms have been elusive. We show that both phenomena can be explained in terms of the population within-class variance of the learned representations, and more specifically, by the progression of neural collapse and its associated time scale. These theoretical findings are supported by our experiments. Beyond the theoretical perspective, our results also provide practical implications: tracking quantities such as the rescaled within-class variance can help determine when continued training will be beneficial, and weight decay can accelerate the transition to this late-phase regime. A natural next step is to extend the time-scale analysis of neural collapse to other architectures beyond MLP or to different initialization methods. Another interesting direction is to analyze the possibility of neural collapse that implicitly arises without weight decay.

## 7 ACKNOWLEDGMENTS

This work was supported by JSPS KAKENHI Grant Number JP24H00709. We thank the reviewers for their constructive comments and the members of our laboratory for helpful discussions.

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

## A  Further Comparison with Related Work

As a concurrent work to ours, we compare our results with Han et al. (2025) in this section. Since the title may appear to contradict our claims at first glance, we clarify that their findings are fully consistent with ours and highlight the novelty of our contribution. Han et al. (2025) discusses grokking in relation to neural collapse, which makes their setting similar to ours. Although the title suggests that neural collapse might not be relevant for generalization, this is not what the paper actually argues. Rather, they state that neural collapse appears earlier than the acquisition of generalization. The emergence of neural collapse does not align with that of generalization, and therefore it does not fully explain generalization ability. In contrast, they observe that flatness correlates more closely with the decrease in test loss and thus appears to offer a better explanation in their experiments. However, this empirical observation is entirely consistent with our analysis. The apparent contradiction arises because Han et al. (2025) measures neural collapse using the neural collapse clustering (NCC) metric, which mixes multiple properties associated with neural collapse. In particular, NCC also captures the separation of class mean representations that naturally occurs through fitting the training set, and therefore it decreases earlier than the test loss. As we emphasize in Remark 4.2, our analysis isolates the contribution of within-class variance or RNC1 score, allowing us to separate out the cause of the generalization and correctly focus on the variance decrease that is actually responsible for the emergence of generalization in grokking. Furthermore, the theoretical analysis in Han et al. (2025) is limited to showing that flatness is partially guaranteed under neural collapse. Their work addresses neither the connection to generalization nor the associated training dynamics. In contrast, as illustrated in Figure 1, our paper clarifies the relationships among several concepts, including information bottleneck, and it further characterizes their training dynamics through a neural-collapse-based analysis. This leads to a unified understanding of the mechanisms underlying grokking.

## B  Proof of Main Results

### B.1  Grokking Results

We first provide the following lemma, which is useful for bounding the tail probability with respect to the variance of the random variable.

**Lemma B.1** (Cantelli's inequality). *Let $X$ be a real-valued random variable with mean $\mathbb{E}[X]$ and variance $\sigma^2$. Then, for any $\lambda > 0$, we have*

$$\Pr\left(X \geq \mathbb{E}[X] + \lambda\right) \leq \frac{\sigma^2}{\sigma^2 + \lambda^2}.$$

Using this lemma, we can prove Theorem 3.2 as follows.

*Proof of Theorem 3.2.* The test error of the model prediction is bounded as follows:

$$\Pr\left(\arg\max_{i \in [K]} \{f(\boldsymbol{x})_i\} \neq y\right) \tag{3}$$

$$= \mathbb{E}_{c \sim P_Y} \mathbb{E}_{\boldsymbol{x} \sim P_{X|Y=c}} \left[\mathbf{1}_{\arg\max_{i \in [K]} \{f(\boldsymbol{x})_i\} \neq c}\right] \tag{4}$$

$$= \sum_{c=1}^{K} p_Y(c) \cdot \Pr\left(\exists k \in [K] \setminus \{c\} \text{ s.t. } \langle g(\boldsymbol{x}), \boldsymbol{w}_k \rangle > \langle g(\boldsymbol{x}), \boldsymbol{w}_c \rangle \mid Y = c\right) \tag{5}$$

$$\leq \sum_{c=1}^{K} p_Y(c) \sum_{k \neq c} \Pr\left(\langle g(\boldsymbol{x}), \boldsymbol{w}_k - \boldsymbol{w}_c \rangle > 0 \mid Y = c\right), \tag{6}$$

where we used union bound in the last argument. From Lemma B.1, the term in the last line can be further bounded as follows:

$$\Pr\left(\langle g(\boldsymbol{x}), \boldsymbol{w}_k - \boldsymbol{w}_c\rangle > 0 \mid Y = c\right) \tag{7}$$

$$= \Pr\left(\langle g(\boldsymbol{x}) - \mathbb{E}_{X|Y=c}\left[g(\boldsymbol{x})\right], \boldsymbol{w}_k - \boldsymbol{w}_c\rangle > \langle \mathbb{E}_{X|Y=c}\left[g(\boldsymbol{x})\right], \boldsymbol{w}_c - \boldsymbol{w}_k\rangle \mid Y = c\right) \tag{8}$$

$$\leq \begin{cases} \frac{(\boldsymbol{w}_c - \boldsymbol{w}_k)^\top \boldsymbol{\Sigma}_c (\boldsymbol{w}_c - \boldsymbol{w}_k)}{(\boldsymbol{w}_c - \boldsymbol{w}_k)^\top \boldsymbol{\Sigma}_c (\boldsymbol{w}_c - \boldsymbol{w}_k) + \langle \mathbb{E}_{X|Y=c}[g(\boldsymbol{x})], \boldsymbol{w}_c - \boldsymbol{w}_k\rangle^2} & \text{if } \langle \mathbb{E}_{X|Y=c}\left[g(\boldsymbol{x})\right], \boldsymbol{w}_c - \boldsymbol{w}_k\rangle > 0, \\ 1 & \text{otherwise,} \end{cases} \tag{9}$$

where $\boldsymbol{\Sigma}_c$ is given by $\boldsymbol{\Sigma}_c = \mathbb{E}_{X|Y=c}\left[\left(g(X) - \mathbb{E}_{X|Y=c}[g(X)]\right)\left(g(X) - \mathbb{E}_{X|Y=c}[g(X)]\right)^\top\right]$. In both cases, Equation (9) can be rewritten as

$$\Pr\left(\langle g(\boldsymbol{x}), \boldsymbol{w}_k - \boldsymbol{w}_c\rangle > 0 \mid Y = c\right) \leq \left(1 + \frac{\max\left\{\langle \mathbb{E}_{X|Y=c}[g(X)], \boldsymbol{w}_c - \boldsymbol{w}_k\rangle, 0\right\}^2}{(\boldsymbol{w}_c - \boldsymbol{w}_k)^\top \boldsymbol{\Sigma}_c (\boldsymbol{w}_c - \boldsymbol{w}_k)}\right)^{-1}. \tag{10}$$

Here, applying the Cauchy-Schwarz inequality to the variance term yields

$$(\boldsymbol{w}_c - \boldsymbol{w}_k)^\top \boldsymbol{\Sigma}_c (\boldsymbol{w}_c - \boldsymbol{w}_k) = \mathbb{E}_{X|Y=c}\left[\langle g(X) - \mathbb{E}_{X|Y=c}\left[g(X)\right], \boldsymbol{w}_c - \boldsymbol{w}_k\rangle^2\right] \tag{11}$$

$$\leq \mathbb{E}_{X|Y=c}\left[\left\|g(X) - \mathbb{E}_{X|Y=c}\left[g(X)\right]\right\|_2^2\right]\|\boldsymbol{w}_c - \boldsymbol{w}_k\|_2^2. \tag{12}$$

By combining Equations (6), (10) and (12), we obtain the desired result:

$$\Pr\left(\arg\max_{i \in [K]}\{f(\boldsymbol{x})_i\} \neq y\right) \leq \sum_{c=1}^{K} p_Y(c) \sum_{k \neq c}\left(1 + \frac{\max\left\{\langle \mathbb{E}_{X|Y=c}[g(X)], \frac{\boldsymbol{w}_c - \boldsymbol{w}_k}{\|\boldsymbol{w}_c - \boldsymbol{w}_k\|_2}\rangle, 0\right\}^2}{\mathbb{E}_{X|Y=c}\left[\left\|g(X) - \mathbb{E}_{X|Y=c}[g(X)]\right\|_2^2\right]}\right)^{-1}$$

$$= \sum_{c=1}^{K} p_Y(c) \sum_{k \neq c}\left(1 + \frac{\max\left\{\langle \mathbb{E}_{X|Y=c}[\tilde{g}(X)], \frac{\boldsymbol{w}_c - \boldsymbol{w}_k}{\|\boldsymbol{w}_c - \boldsymbol{w}_k\|_2}\rangle, 0\right\}^2}{\mathbb{E}_{X|Y=c}\left[\left\|\tilde{g}(X) - \mathbb{E}_{X|Y=c}[\tilde{g}(X)]\right\|_2^2\right]}\right)^{-1},$$

where the last line follows from Definition 3.1 and dividing both the numerator and denominator by $B_g^2 = \sup_{\boldsymbol{x} \in \mathcal{X}} \|g(\boldsymbol{x})\|_2^2$. $\qquad\square$

## B.2 IB Dynamics

### B.2.1 Proof of Proposition 3.3

*Proof of Proposition 3.3.* By definition, $Z$ is obtained by adding noise to $g(X)$, so the Markov chain $Y \to X \to g(X) \to Z$ holds. The first inequality $I(Z; X) \geq I(Z; Y)$ follows from the Markov chain $Y \to X \to Z$ and the data processing inequality (DPI). For the second inequality, we use a variational approach. From a definition of MI, we have

$$I(Z; Y) = \int dy\, d\boldsymbol{z}\, p(y, \boldsymbol{z}) \log \frac{p(y, \boldsymbol{z})}{p(y)p(\boldsymbol{z})} = \int dy\, d\boldsymbol{z}\, p(y, \boldsymbol{z}) \log \frac{p(y|\boldsymbol{z})}{p(y)}. \tag{13}$$

Here, we introduce a variational approximation $q(y|\boldsymbol{z})$ for the conditional distribution $p(y|\boldsymbol{z})$. From the non-negativity of the KL divergence, we have

$$I(Z; Y) \geq \int dy\, d\boldsymbol{z}\, p(y, \boldsymbol{z}) \log \frac{q(y|\boldsymbol{z})}{p(y)} = \int dy\, d\boldsymbol{z}\, p(y, \boldsymbol{z}) \log q(y|\boldsymbol{z}) + H(Y). \tag{14}$$

We model the variational approximation as a softmax function with respect to the last layer output $\boldsymbol{W}\boldsymbol{z}$, i.e., $q(y|\boldsymbol{z}) = \exp\left((\boldsymbol{W}\boldsymbol{z})_y\right) / \sum_{c=1}^{K} \exp\left((\boldsymbol{W}\boldsymbol{z})_c\right)$, leading to

$$I(Z; Y) \geq -\mathbb{E}_{(y, \boldsymbol{z}) \sim (Y, Z)}\left[\ell_{CE}(y, \boldsymbol{z})\right] + H(Y). \tag{15}$$

Since the entropy of the target $Y$ is a constant, we conclude the desired inequality. $\qquad\square$

### B.2.2 PROOF OF THEOREM 3.4

Before moving on to the proof of Theorem 3.4, we provide the following lemma, which states that the Gaussian distribution maximizes the entropy among all distributions with the same covariance.

**Lemma B.2** (Cover & Thomas (2006), Theorem 8.6.5). *Let the random vector $X \in \mathbb{R}^d$ have zero mean and covariance matrix $\mathbf{\Sigma} = \mathbb{E}[XX^\top]$. Then, we have $h(X) \leq \frac{1}{2} \log \left\{ (2\pi e)^d \det(\mathbf{\Sigma}) \right\}$, with equality if and only if $X \sim N(\mathbf{0}, \mathbf{\Sigma})$.*

With this lemma, we show the proof of Theorem 3.4.

*Proof of Theorem 3.4.* Using the Markov chain $Y \to X \to Z$ and the chain rule of MI, we have

$$I(Z; X) = I(Z; X, Y) = I(Z; Y) + I(Z; X|Y), \tag{16}$$

leading to the first equality. Rewriting the conditional MI with the differential entropies, we have

$$I(Z; X|Y) = h(Z|Y) - h(Z|X, Y) = h(Z|Y) - h(Z|X), \tag{17}$$

which again follows from the Markov chain. For the second term, we use the differential entropy of Gaussian distribution (Cover & Thomas, 2006, Theorem 8.4.1), leading to

$$h(Z|X) = h\left(g(X) + B_g E|X\right) = h(B_g E|X) = \frac{d_{\text{rep}}}{2}\left(1 + \log(2\pi B_g^2 \sigma^2)\right). \tag{18}$$

The conditional covariance matrix of $Z$ given $Y = y$ is given by

$$\mathbf{\Sigma}_{Z|Y=y} = \text{Cov}_{X|Y=y}\left[g(X)\right] + B_g^2 \sigma^2 \mathbf{I}_{d_{\text{rep}}}. \tag{19}$$

Then, from Lemma B.2, the first term in Equation (17) can be computed as follows:

$$h(Z|Y)$$
$$= \mathbb{E}_{y \sim Y}\left[h(Z|Y = y)\right] \tag{20}$$

$$\leq \frac{1}{2} \mathbb{E}_{y \sim Y}\left[\log\left\{(2\pi e)^{d_{\text{rep}}} \det(\mathbf{\Sigma}_{Z|Y=y})\right\}\right] \tag{21}$$

$$= \frac{1}{2}\left(d_{\text{rep}}\left(1 + \log(2\pi B_g^2 \sigma^2)\right) + \mathbb{E}_{y \sim Y}\left[\log \det\left(\frac{\text{Cov}_{X|Y=y}\left[g(X)\right]}{B_g^2 \sigma^2} + \mathbf{I}_{d_{\text{rep}}}\right)\right]\right) \tag{22}$$

$$\leq \frac{1}{2}\left(d_{\text{rep}}\left(1 + \log(2\pi B_g^2 \sigma^2)\right) + \mathbb{E}_{y \sim Y}\left[\text{Tr}\left(\frac{\text{Cov}_{X|Y=y}\left[g(X)\right]}{B_g^2 \sigma^2}\right)\right]\right), \tag{23}$$

where the last inequality follows from the fact that $\log \det(\mathbf{A} + \mathbf{I}) \leq \text{Tr}(\mathbf{A})$ for any positive semidefinite matrix $\mathbf{A}$. By putting Equation (18) and Equation (23) into Equation (17), we have

$$I(Z; X|Y) \leq \frac{1}{2B_g^2 \sigma^2} \mathbb{E}_{y \sim Y}\left[\text{Tr}\left(\text{Cov}_{X|Y=y}\left[g(X)\right]\right)\right] \tag{24}$$

$$= \frac{1}{2\sigma^2} \mathbb{E}_{(\boldsymbol{x}, y) \sim (X, Y)}\left[\left\|\tilde{g}(\boldsymbol{x}) - \mathbb{E}_{\boldsymbol{x} \sim X|Y=y}\left[\tilde{g}(\boldsymbol{x})\right]\right\|_2^2\right], \tag{25}$$

which concludes the proof. $\qquad\square$

### B.2.3 REMARK ON UPPER-BOUND TIGHTNESS

In Theorem 3.4, the redundant information $I(Z; X \mid Y)$ is upper-bounded with a population within-class variance. In this section, we discuss how informative this upper bound is.

From the proof of Theorem 3.4, we obtain the following tight bound on $I(Z; X \mid Y)$:

$$I(Z; X \mid Y) \leq \frac{1}{2} \mathbb{E}_{y \sim Y}\left[\log \det\left(\frac{\text{Cov}_{X|Y=y}[\tilde{g}(X)]}{\sigma^2} + \mathbf{I}\right)\right]. \tag{26}$$

This bound is derived from Lemma B.2, and equality holds when $Z \mid Y = y$ follows the Gaussian distribution. Therefore, this upper bound is tight. In the subsequent step of the proof, we used the inequality $\log \det(\mathbf{A} + \mathbf{I}) \leq \text{Tr}(\mathbf{A})$. Since the equality does not hold when $\mathbf{A}$ is positive definite, the resulting bound in Theorem 3.4 is not tight. However, regarding the within-class variance term appearing in the theorem, we can show that reducing the variance in each coordinate always decreases both the tight bound above and the upper bound in Theorem 3.4

**Proposition B.3.** *Reducing the population within-class variance in any single coordinate, i.e.,* $\mathbb{E}_{X|Y=y}\left[\left(\tilde{g}(X) - \mathbb{E}_{X|Y=y}[\tilde{g}(X)]\right)_i^2\right]$ *for any* $i \in [d_{rep}]$ *and* $y \in [K]$, *strictly decreases both the upper bound in Theorem 3.4 and the tight upper bound on* $I(Z; X \mid Y)$ *in Equation (26).*

*Proof of Proposition B.3.* For the upper bound in Theorem 3.4, since it can be written as

$$\sum_{i \in [d_{rep}]} \mathbb{E}_Y \mathbb{E}_{X|Y} \left[\left(\tilde{g}(X) - \mathbb{E}_{X|Y}[\tilde{g}(X)]\right)_i^2\right], \tag{27}$$

it is obvious that decreasing any summand decreases the upper bound. We now show that the tight upper bound in Equation (26) also decreases under such a perturbation. Let the amount of the variance reduction along coordinate $i$ be $\delta > 0$. For notational simplicity, we denote the original covariance matrix by $\boldsymbol{K} = \mathrm{Cov}_{X|Y}[\tilde{g}(X)]$ and the perturbed covariance by $\boldsymbol{K}' = \boldsymbol{K} - \delta \boldsymbol{e}_i \boldsymbol{e}_i^\top$. Then, the value of the upper bound is bounded below as

$$\frac{1}{2} \mathbb{E}_Y \left[\log \det \left(\frac{\boldsymbol{K}' + \delta \boldsymbol{e}_i \boldsymbol{e}_i^\top}{\sigma^2} + \boldsymbol{I}\right)\right] \tag{28}$$

$$= \frac{1}{2} \mathbb{E}_Y \left[\log \left\{\left(1 + \frac{\delta}{\sigma^2} \boldsymbol{e}_i^\top \left(\frac{\boldsymbol{K}'}{\sigma^2} + \boldsymbol{I}\right)^{-1} \boldsymbol{e}_i\right) \det \left(\frac{\boldsymbol{K}'}{\sigma^2} + \boldsymbol{I}\right)\right\}\right] \tag{29}$$

$$> \frac{1}{2} \mathbb{E}_Y \left[\log \det \left(\frac{\boldsymbol{K}'}{\sigma^2} + \boldsymbol{I}\right)\right], \tag{30}$$

where the equality holds from the matrix determinant lemma, and the inequality follows from the fact that the diagonal element of the inverse of a positive definite matrix is always positive. Since the right-hand side is exactly the upper bound in Equation (26) evaluated at the perturbed covariance, this completes the proof. $\square$

## B.3 CONCENTRATION OF WITHIN-CLASS VARIANCE

The analysis in Section 4.1 is carried out using Rademacher complexity, which is a standard tool for establishing uniform convergence and deriving generalization bounds. For a real-valued function class $\mathcal{F}$ and a fixed training set $S$, the empirical Rademacher complexity $\widehat{\mathfrak{R}}_n(\mathcal{F})$ is defined as $\mathbb{E}_{\boldsymbol{\epsilon}}\left[\sup_{f \in \mathcal{F}} \frac{1}{n} \sum_{i=1}^n \epsilon_i f(\boldsymbol{x}_i)\right]$, where $\epsilon_i$ are independently sampled from $\mathrm{Unif}(\{\pm 1\})$.

Theorem 4.1 is inspired by Galanti et al. (2022, Proposition 3). In contrast to their result, the following theorem addresses the three novel aspects: (i) a uniform convergence result for within-class variance, (ii) a refined design of the failure probabilities and the union bound, and (iii) a precise and tight upper bound on Rademacher complexity.

*Proof of Theorem 4.1.* We first define the function classes of DNNs as follows:

$$\mathcal{G} = \left\{g : \mathbb{R}^d \to \mathbb{R}^{d_{\mathrm{rep}}} \mid g(\boldsymbol{x}) = \sigma\left(\boldsymbol{W}_{L-1}\sigma\left(\cdots\sigma\left(\boldsymbol{W}_1 \boldsymbol{x}\right)\right)\right), \boldsymbol{W}_\ell \in \mathbb{R}^{d_\ell \times d_{\ell-1}}, \ell \in [L-1]\right\},$$

$$\mathcal{G}_{s,t} = \left\{g \in \mathcal{G} \;\middle|\; \Pi(g) = \prod_{\ell=1}^{L-1} \|\boldsymbol{W}_\ell\|_2 \leq 2^s, \; \Lambda(g) = \left(\sum_{\ell=1}^{L-1} \left(\frac{\|\boldsymbol{W}_\ell\|_{2,1}}{\|\boldsymbol{W}_\ell\|_2}\right)^{2/3}\right)^{3/2} \leq 2^t\right\},$$

for $s, t \in \mathbb{N}$. For each of these classes, we define corresponding rescaled classes as follows:

$$\tilde{\mathcal{G}} = \left\{\tilde{g} \mid \tilde{g}(\boldsymbol{x}) = \frac{g(\boldsymbol{x})}{\sup_{\boldsymbol{x} \in \mathcal{X}} \|g(\boldsymbol{x})\|_2}, g \in \mathcal{G}\right\}, \quad \tilde{\mathcal{G}}_{s,t} = \left\{\tilde{g} \mid \tilde{g}(\boldsymbol{x}) = \frac{g(\boldsymbol{x})}{\sup_{\boldsymbol{x} \in \mathcal{X}} \|g(\boldsymbol{x})\|_2}, g \in \mathcal{G}_{s,t}\right\}.$$

Then, we have $\mathcal{G} = \bigcup_{s,t} \mathcal{G}_{s,t}$ and $\tilde{\mathcal{G}} = \bigcup_{s,t} \tilde{\mathcal{G}}_{s,t}$.

We first fix $c \in [K]$ and $s, t \in \mathbb{N}$. For any $\tilde{g} \in \tilde{\mathcal{G}}_{s,t}$, we have

$$
\mathbb{E}_{X|Y=c}\left[\left\|\tilde{g}(X) - \mathbb{E}_{X|Y=c}\left[\tilde{g}(X)\right]\right\|_2^2\right]
$$

$$
= \mathbb{E}_{X|Y=c}\left[\left\|\tilde{g}(X) - \frac{1}{n_c}\sum_{j \in S_c}\tilde{g}(\boldsymbol{x}_j)\right\|_2^2 - \left\|\frac{1}{n_c}\sum_{j \in S_c}\tilde{g}(\boldsymbol{x}_j) - \mathbb{E}_{X|Y=c}\left[\tilde{g}(X)\right]\right\|_2^2\right] \quad (31)
$$

$$
\leq \mathbb{E}_{X|Y=c}\left[\left\|\tilde{g}(X) - \frac{1}{n_c}\sum_{j \in S_c}\tilde{g}(\boldsymbol{x}_j)\right\|_2^2\right]. \quad (32)
$$

In the following, we will analyze the gap between Equation (32) and its empirical counterpart. For any fixed $\tilde{g} \in \tilde{\mathcal{G}}_{s,t}$, we define a function $h : \mathbb{R}^{d_{\mathrm{rep}}} \to \mathbb{R}$ as $h(\boldsymbol{z}) = \left\|\boldsymbol{z} - \frac{1}{n_c}\sum_{j \in S_c}\tilde{g}(\boldsymbol{x}_j)\right\|_2^2$. Since the output of $\tilde{g} \in \tilde{\mathcal{G}}_{s,t}$ is rescaled, we have $\|\tilde{g}(\boldsymbol{x})\|_2 \leq 1$ for all $\boldsymbol{x} \in \mathcal{X}$, and the output of $h \circ \tilde{g}$ is bounded with $4$. By normalizing the output with this value to apply Mohri et al. (2018, Theorem 3.3), with probability at least $1 - \delta_{s,t}$, we have

$$
\left|\mathbb{E}_{X|Y=c}\left[\left\|\tilde{g}(X) - \frac{1}{n_c}\sum_{j \in S_c}\tilde{g}(\boldsymbol{x}_j)\right\|_2^2\right] - \frac{1}{n_c}\sum_{i \in S_c}\left\|\tilde{g}(\boldsymbol{x}_i) - \frac{1}{n_c}\sum_{j \in S_c}\tilde{g}(\boldsymbol{x}_j)\right\|_2^2\right|
$$

$$
\leq 2\widehat{\mathfrak{R}}_{n_c}(h \circ \tilde{\mathcal{G}}_{s,t}) + 4 \cdot 3\sqrt{\frac{\log(2/\delta_{s,t})}{2n_c}}. \quad (33)
$$

By choosing the failure probabilities as $\delta_{s,t} = \delta/(Kst(s+1)(t+1))$ and applying the union bound, since we have $\sum_{s,t=1}^{\infty}\delta_{s,t} = \delta/K$, the above inequality holds with probability at least $1 - \delta$ for all $c \in [K]$, $s, t \in \mathbb{N}$ and $\tilde{g} \in \tilde{\mathcal{G}}_{s,t}$.

Next, we analyze the Rademacher complexity term in Equation (33) using a covering number argument. For a set $U$, we define its covering number $\mathcal{N}(U, \epsilon, \|\|)$ as the minimal cardinality of a subset $V \subseteq U$ such that, for every $u \in U$, there exists $v \in V$ satisfying $\|u - v\| \leq \epsilon$. Here, for the real-valued function class $\mathcal{F}$, we define its restriction to the training data points as

$$
\mathcal{F}_{|S_c} = \left\{\boldsymbol{x} \mapsto (f(\boldsymbol{x}_1), \ldots, f(\boldsymbol{x}_{n_c}))^\top \in \mathbb{R}^{n_c} \mid f \in \mathcal{F}\right\}. \quad (34)
$$

As discussed earlier, both the domain of the function $h$ and the average $\sum_{j \in S_c}\tilde{g}(\boldsymbol{x}_j)/n_c$ are restricted to the unit $\ell_2$-ball. Thus, the Lipschitz constant of $h$ is given by $h_{\mathrm{Lip}} = 4$. By Bartlett et al. (2017, Theorem 3.3) and the definition of $\tilde{\mathcal{G}}_{s,t}$, we have

$$
\sqrt{\log \mathcal{N}\left(\left(h \circ \tilde{\mathcal{G}}_{s,t}\right)_{|S_c}, \epsilon, \|\cdot\|_2\right)} \leq \frac{B_x\sqrt{n_c\log(2d_{\max})}}{\epsilon} \cdot \frac{4}{B_g} \cdot 2^s 2^t. \quad (35)
$$

Using Bartlett et al. (2017, Lemma A.5), modified so that the range is extended from $[0, 1]$ to $[0, 4]$, and applying Equation (35), we have

$$
\widehat{\mathfrak{R}}_{n_c}(h \circ \mathcal{G}_{s,t}) \leq \inf_{\alpha > 0}\left(\frac{4\alpha}{\sqrt{n_c}} + \frac{12}{n_c}\int_\alpha^{4\sqrt{n_c}}\sqrt{\log \mathcal{N}\left(\left(h \circ \tilde{\mathcal{G}}_{s,t}\right)_{|S_c}, \epsilon, \|\cdot\|_2\right)}d\epsilon\right) \quad (36)
$$

$$
\leq \inf_{\alpha > 0}\left(\frac{4\alpha}{\sqrt{n_c}} + \frac{12}{n_c}\log\left(\frac{4\sqrt{n_c}}{\alpha}\right)\sqrt{n_c\log(2d_{\max})}\frac{B_x}{B_g}2^{s+t+2}\right) \quad (37)
$$

$$
\leq \frac{16}{n_c} + \frac{12 \cdot 2^{s+t+2}B_x}{\sqrt{n_c}B_g}\log(n_c)\sqrt{\log(2d_{\max})}, \quad (38)
$$

where the last line follows by taking $\alpha = 4/\sqrt{n_c}$. Substituting this into Equation (33), we have

$$
\left| \mathbb{E}_{X|Y=c} \left[ \left\| \tilde{g}(X) - \frac{1}{n_c} \sum_{j \in S_c} \tilde{g}(\boldsymbol{x}_j) \right\|_2^2 \right] - \frac{1}{n_c} \sum_{i \in S_c} \left\| \tilde{g}(\boldsymbol{x}_i) - \frac{1}{n_c} \sum_{j \in S_c} \tilde{g}(\boldsymbol{x}_j) \right\|_2^2 \right|
$$
$$
\leq \frac{32}{n_c} + \frac{12 \cdot 2^{s+t+3} B_x}{\sqrt{n_c} B_g} \log(n_c) \sqrt{\log(2 d_{\max})} + 12 \sqrt{\frac{\log(2Kst(s+1)(t+1)/\delta)}{2 n_c}}. \tag{39}
$$

For any $\tilde{g} \in \tilde{\mathcal{G}}$, let $s := \lfloor \log_2(2\Pi(g)) \rfloor$ and $t := \lfloor \log_2(2\Lambda(g)) \rfloor$, so that $\tilde{g} \in \tilde{\mathcal{G}}_{s,t}$ with $2^{s-1} \leq \Pi(g) \leq 2^s$ and $2^{t-1} \leq \Lambda(g) \leq 2^t$. From Equation (39), it follows that with probability at least $1 - \delta$, we have

$$
\left| \mathbb{E}_{X|Y=c} \left[ \left\| \tilde{g}(X) - \frac{1}{n_c} \sum_{j \in S_c} \tilde{g}(\boldsymbol{x}_j) \right\|_2^2 \right] - \frac{1}{n_c} \sum_{i \in S_c} \left\| \tilde{g}(\boldsymbol{x}_i) - \frac{1}{n_c} \sum_{j \in S_c} \tilde{g}(\boldsymbol{x}_j) \right\|_2^2 \right|
$$
$$
\leq O\left( \frac{1}{\sqrt{n_c}} \left[ \frac{1}{\sqrt{n_c}} + \frac{\Pi(g) B_x}{B_g} \log(n_c) \sqrt{\log(d_{\max})} \Lambda(g) + \sqrt{\log(K/\delta) + \log\log(\Pi(g)\Lambda(g))} \right] \right),
$$

which completes the proof. $\qquad \square$

## B.4 Neural Collapse Dynamics

### B.4.1 Preliminaries

We first provide the formal assumptions used in Theorem 4.3, adopted from Jacot et al. (2025). A detailed discussion of their validity and typicality is given in Remark B.6.

**Assumption B.3** (Pyramidal network). Let $d_1 \geq N$ and $d_\ell \geq d_{\ell+1}$ for all $\ell \in \{2, \ldots, L-1\}$.

**Assumption B.4** (Smooth activation). Let $\gamma \in (0,1)$ and $\beta \geq 1$. Suppose $\sigma$ satisfies: (i) $\sigma'(x) \in [\gamma, 1]$ for all $x \in \mathbb{R}$; (ii) $|\sigma(x)| \leq |x|$ for every $x \in \mathbb{R}$; and (iii) $\sigma'$ is $\beta$-Lipschitz continuous.

**Assumption B.5** (Initialization). Let $\lambda_\ell = \sigma_{\min}(\boldsymbol{W}_\ell(0))$ and $\lambda_F = \sigma_{\min}(\sigma(\boldsymbol{W}_1(0)\boldsymbol{X}))$, where $\sigma_{\min}(\cdot)$ denotes the smallest singular value of a matrix. Suppose that we have

$$
\lambda_F \prod_{\ell=3}^{L} \lambda_\ell \min\left(\lambda_F, \min_{\ell \in \{3,\ldots,L\}} \lambda_\ell\right) \geq 8\gamma \sqrt{\left(\frac{2}{\gamma}\right)^L \widehat{\mathcal{L}}_0(\boldsymbol{\theta}(0))}.
$$

In the following, for notational convenience, we denote by $\boldsymbol{Z}_\ell \in \mathbb{R}^{d_\ell \times N}$ the output of the $\ell$-th layer for the entire training set. Specifically, $\boldsymbol{Z}_0 = \boldsymbol{X}_0$, $\boldsymbol{Z}_\ell = \sigma(\boldsymbol{W}_\ell \boldsymbol{Z}_{\ell-1})$ for $\ell \in [L-2]$, and $\boldsymbol{Z}_\ell = \boldsymbol{W}_\ell \boldsymbol{Z}_{\ell-1}$ for $\ell \in \{L-1, L\}$. Additionally, we denote $\bar{\lambda}_\ell = \|\boldsymbol{W}_\ell(0)\|_2 + \min_{\ell \in \{3,\ldots,L\}} \lambda_\ell$ for $\ell \in [L]$. We denote by $\sigma_i(\cdot)$ the $i$-th largest singular value of a given matrix.

**Remark B.6** (Validity of the Assumptions). The derivation of neural collapse from gradient descent, rather than from the unconstrained feature model (UFM), was first established in Jacot et al. (2025), and Theorem 4.3 adopts the same assumptions. To analyze the convergence of the DNN training loss, one typically imposes conditions ensuring a positive lower bound on the Jacobian with respect to the parameters. There are multiple well-established ways to guarantee this, including width requirements across all layers (Du et al., 2019; Allen-Zhu et al., 2019; Zou & Gu, 2019) and other pyramidal-topology-based conditions with mild width assumptions (Nguyen & Mondelli, 2020; Bombari et al., 2022; Karhadkar et al., 2024). Our Assumptions B.3 to B.5 can be replaced by any of these alternatives. For example, Allen-Zhu et al. (2019); Zou & Gu (2019) analyze ReLU networks instead of smooth activations but require all layers to be sufficiently wide. In contrast, the pyramidal topology assumption (Assumption B.3) removes the need for width assumptions on every layer and replaces them with a more realistic setting: a wide first layer followed by a narrowing architecture. The smooth-activation assumption (Assumption B.4) is widely used in convergence analysis and appears in independent work such as Nguyen & Mondelli (2020); Bombari et al. (2022); Liu et al. (2022a); Frei et al. (2022). Smooth leaky ReLU satisfies this assumption and can approximate ReLU arbitrarily well for suitable choices of $\gamma$ and $\beta$. Assumption B.5 concerns initialization,

and it can be satisfied by choosing the second-layer scale sufficiently small. Moreover, by relaxing Assumption B.3 from $d_1 \geq N$ to $d_1 = \Omega(N)$, Assumption B.5 can be shown to hold under standard He/LeCun initialization (LeCun et al., 2002; He et al., 2015), which follows directly from Appendix C of Nguyen & Mondelli (2020). Finally, the use of squared loss is standard in this line of work: it is used both in the gradient-descent NTK-style analyses (Du et al., 2019; Allen-Zhu et al., 2019; Zou & Gu, 2019; Nguyen & Mondelli, 2020; Jacot et al., 2025) and in theoretical neural collapse literature (Han et al., 2022; Zhou et al., 2022a; Súkeník et al., 2023; 2024). As noted in Jacot et al. (2025), once the training set is interpolated, the second part of the analysis (the time step $\tau_2$ in Theorem 4.3) showing the emergence of neural collapse does not rely on these assumptions. We note that none of our results except Theorem 4.3 depend on Assumptions B.3 to B.5.

### B.4.2 PROOF OF THEOREM 4.3

We now provide a proof of Theorem 4.3. The argument follows that of Jacot et al. (2025, Theorem B.2). Specifically, for the convergence of the training loss, we employ the standard Polyak-Łojasiewicz (PL) condition to establish linear convergence. On the other hand, the progression of NC1 and RNC1 is explained through the development of weight balancedness, i.e., $\boldsymbol{W}_L^\top \boldsymbol{W}_L \approx \boldsymbol{W}_{L-1} \boldsymbol{W}_{L-1}^\top$. The key idea is that once the network output interpolates the target sufficiently well, which results in a small training loss, the weight balancedness ensures that the degree of interpolation is inherited by the output of the preceding layer, i.e., the representation space.

Since our primary focus is on the convergence of the RNC1 metric, we begin by establishing the necessary result.

**Proposition B.7** (Theorem B.1 of Jacot et al. (2025)). *If the network satisfies (i) approximate interpolation, i.e., $\|f(\boldsymbol{X}) - \boldsymbol{Y}\|_F \leq \xi_1$, (ii) approximate balancedness, i.e., $\|\boldsymbol{W}_L^\top \boldsymbol{W}_L - \boldsymbol{W}_{L-1} \boldsymbol{W}_{L-1}^\top\|_2 \leq \xi_2$, and (iii) bounded representations and weights, i.e., $\|\boldsymbol{Z}_{L-2}\|_2 \leq r$, $\|\boldsymbol{Z}_{L-1}\|_2 \leq r$, and $\|\boldsymbol{W}_\ell\|_2 \leq r$ for $\ell \in [L]$, then if $\xi_1 \leq \min\left\{\sigma_K(\boldsymbol{Y}), \sqrt{\frac{(K-1)N}{4K}}\right\}$, we have*

$$\mathrm{Tr}\left(\boldsymbol{\Sigma}_W\right) \leq \frac{r^2}{N}\left(\frac{\xi_1}{\sigma_K(\boldsymbol{Y}) - \xi_1} + \sqrt{d_{L-1}\xi_2}\right)^2.$$

**Corollary B.8.** *Under the conditions of Proposition B.7, if $\xi_1 \leq \frac{1}{2}\min_{c\in[K]}\sqrt{n_c}$, we have*

$$\mathrm{RNC1} \leq \frac{r^4}{N\left(1 - \frac{\xi_1}{\sqrt{N}}\right)^2}\left(\frac{\xi_1}{\min_{c\in[K]}\sqrt{n_c} - \xi_1} + \sqrt{d_{L-1}\xi_2}\right)^2.$$

*Proof of Corollary B.8.* By the definition of RNC1, it suffices to bound $\mathrm{Tr}(\boldsymbol{\Sigma}_W)$ from above and $B_g$ from below. Substituting $\sigma_K(\boldsymbol{Y}) = \min_{c\in[K]}\sqrt{n_c}$, which follows from $\boldsymbol{Y}\boldsymbol{Y}^\top = \mathrm{diag}(n_1, \ldots, n_K)$, into Proposition B.7, we have

$$\mathrm{Tr}(\boldsymbol{\Sigma}_W) \leq \frac{r^2}{N}\left(\frac{\xi_1}{\min_{c\in[K]}\sqrt{n_c} - \xi_1} + \sqrt{d_{L-1}\xi_2}\right)^2. \tag{40}$$

For the lower bound on $B_g$, we show the existence of a training example whose norm can be bounded from below. From the condition $\|f(\boldsymbol{X}) - \boldsymbol{Y}\|_F \leq \xi_1$, there exists $i \in [N]$ such that $\|f(\boldsymbol{x}_i) - \boldsymbol{y}_i\|_2 \leq \xi_1/\sqrt{N}$, which implies $\|\boldsymbol{y}_i\|_2 - \xi_1/\sqrt{N} \leq \|f(\boldsymbol{x}_i)\|_2$. Since $\boldsymbol{y}_i$ is a one-hot vector and by $\|\boldsymbol{W}_L\|_2 \leq r$, we have

$$\frac{1}{r}\left(1 - \frac{\xi_1}{\sqrt{N}}\right) \leq \|g(\boldsymbol{x}_i)\|_2 \leq B_g. \tag{41}$$

Combining Equations (40) and (41) yields the desired upper bound. Finally, regarding the condition on $\xi_1$, we used that $\sigma_K(\boldsymbol{Y}) = \min_{c\in[K]}\sqrt{n_c} \leq \sqrt{N/K}$ and $K \geq 2$.

$\square$

**Proposition B.9.** *Suppose that the network $f$ satisfies Assumptions B.3 to B.5 and that the input domain $\mathcal{X}$ is bounded, i.e., $\|\boldsymbol{x}\|_2 \leq B_x$ for all $\boldsymbol{x} \in \mathcal{X}$. Fix $0 < \epsilon_1 < \frac{1}{2\sqrt{2}}\min_{c\in[K]}\sqrt{n_c}$ and $\epsilon_2 > 0$.*

*Let weight decay parameter $\lambda$ and step size $\eta$ satisfy the following:*

$$\lambda \leq \min \left\{ 2^{-(L-3)} \gamma^{L-2} \lambda_F \prod_{\ell=3}^{L} \lambda_\ell, \; \frac{2\widehat{\mathcal{L}}_0(\boldsymbol{\theta}(0))}{\|\boldsymbol{\theta}(0)\|_2^2}, \; \frac{\epsilon_1^2}{18 \left( \|\boldsymbol{\theta}(0)\|_2 + \lambda_F/2 \right)^2} \right\},$$

$$\eta \leq \min \left\{ \frac{1}{2\beta_1}, \; \frac{1}{5N\beta B_x^3 \max\left\{1, (4m_\lambda)^{3L/2}\right\} L^{5/2}}, \; \frac{1}{2\lambda}, \; \frac{\epsilon_2}{4 \left(4m_\lambda\right)^L \|\boldsymbol{X}\|_2^2} \right\},$$

*where $\beta_1 = 5N\beta B_x^3 \left( \prod_{\ell=1}^{L} \max\{1, \bar{\lambda}_\ell\} \right)^3 L^{5/2}$ and $m_\lambda = \left(1 + \sqrt{4\lambda/\alpha}\right)^2 \left(\|\boldsymbol{\theta}(0)\|_2 + r_0\right)^2$, with $r_0 = \frac{1}{2} \min\left\{\lambda_F, \min_{\ell \in \{3,\dots,L\}} \lambda_\ell\right\}$ and $\alpha = 2^{-(L-3)} \gamma^{L-2} \lambda_F \prod_{\ell=3}^{L} \lambda_\ell$. Then, there exist time steps*

$$\tau_1' \leq \left\lceil \frac{\log \frac{\epsilon_1}{\widehat{\mathcal{L}}_\lambda(\boldsymbol{\theta}(0)) - \lambda m_\lambda}}{\log \left(1 - \eta\frac{\alpha}{8}\right)} \right\rceil, \; \tau_1 \leq \left\lceil \frac{\log \frac{\lambda m_\lambda}{\widehat{\mathcal{L}}_\lambda(\boldsymbol{\theta}(0)) - \lambda m_\lambda}}{\log \left(1 - \eta\frac{\alpha}{8}\right)} \right\rceil, \; \tau_2 \leq \tau_1 + \left\lceil \frac{\log \left(\frac{\epsilon_2}{8m_\lambda}\right)}{\log \left(1 - \eta\lambda\right)} \right\rceil,$$

*such that for any time step $\tau \geq \tau_1'$, we have*

$$\widehat{\mathcal{L}}_\lambda(\boldsymbol{\theta}(\tau)) \leq \epsilon_1^2,$$

*and for any time step $\tau \geq \tau_2$, we have*

$$\text{RNC1}(\tau) \leq \frac{r^4}{N \left(1 - \sqrt{\frac{2}{N}\epsilon_1}\right)^2} \left( \frac{\sqrt{2}\epsilon_1}{\min_{c \in [K]} \sqrt{n_c} - \sqrt{2}\epsilon_1} + \sqrt{d_{L-1}\epsilon_2} \right)^2,$$

*where*

$$r = \max \left\{ 2\sqrt{m_\lambda}, \; (2\sqrt{m_\lambda})^{L-2} \|\boldsymbol{X}\|_2, \; (2\sqrt{m_\lambda})^{L-1} \|\boldsymbol{X}\|_2 \right\}.$$

*Proof of Proposition B.9.* The proof follows the same argument as in the proof of Jacot et al. (2025, Theorem B.2). Under Assumptions B.3 to B.5, the PL property of the unregularized loss $\widehat{\mathcal{L}}_0(\boldsymbol{\theta})$ is established. While the presence of a weight-decay term slightly shifts the convergence point, the PL condition still holds for the regularized loss $\widehat{\mathcal{L}}_\lambda(\boldsymbol{\theta})$. From the PL condition of $\widehat{\mathcal{L}}_\lambda(\boldsymbol{\theta})$ around the initialization, there exists a time step

$$\tau_1 \leq \left\lceil \frac{\log \frac{\lambda m_\lambda}{\widehat{\mathcal{L}}_\lambda(\boldsymbol{\theta}(0)) - \lambda m_\lambda}}{\log \left(1 - \eta\frac{\alpha}{8}\right)} \right\rceil \tag{42}$$

such that $\widehat{\mathcal{L}}_\lambda(\tau_1) \leq 2\lambda m_\lambda < \epsilon_1^2$. Here, note that if the goal is only to ensure that $\widehat{\mathcal{L}}_\lambda(\tau_1') < \epsilon_1^2$, the choice of $\tau_1'$ in the statement suffices.

If the regularized loss is smooth, meaning that $\nabla \widehat{\mathcal{L}}_\lambda(\boldsymbol{\theta})$ is $\beta_2$-Lipschitz continuous and if the learning rate is chosen no larger than $1/\beta_2$, then the loss remains non-increasing for all $\tau \geq \tau_1$. To evaluate $\beta_2$, it is necessary to bound the parameter norm from above. To obtain the norm bound that is independent of $\lambda$, we use $2\lambda m_\lambda$ instead of $\epsilon_1^2$ in the original proof and evaluate as:

$$\frac{\lambda}{2} \|\boldsymbol{\theta}(\tau)\|_2^2 \leq \widehat{\mathcal{L}}_\lambda(\boldsymbol{\theta}(\tau)) \leq 2\lambda m_\lambda, \tag{43}$$

which gives $\|\boldsymbol{\theta}(\tau)\|_2 \leq 2\sqrt{m_\lambda}$. Therefore, Jacot et al. (2025, Lemma C.1) provides $\beta_2 = 5N\beta B_x^3 \max\left\{1, (4m_\lambda)^{3L/2}\right\} L^{5/2}$. Under the learning rate specified in the assumption, the inequality $\widehat{\mathcal{L}}_\lambda(\boldsymbol{\theta}(\tau)) \leq \epsilon_1^2$ holds for all time steps beyond $\tau_1$ (or $\tau_1'$).

The remainder of the proof, including weight balancedness, proceeds as in the original argument by combining the above weight norm bound with the interpolation bound $\|f_\tau(\boldsymbol{X}) - \boldsymbol{Y}\|_F \leq \sqrt{2}\epsilon_1$ for all $\tau \geq \tau_1$. Specifically, the weight balancedness $\boldsymbol{W}_L(\tau)^\top \boldsymbol{W}_L(\tau) - \boldsymbol{W}_{L-1}(\tau)\boldsymbol{W}_{L-1}(\tau)^\top$ is shown

to converge under a sufficiently small learning rate, with the argument based on the analysis of its one-step update. In summary, there exists a time step $\tau_2$ such that

$$\tau_2 \leq \tau_1 + \left\lceil \frac{\log\left(\frac{\epsilon_2}{8m_\lambda}\right)}{\log\left(1 - \eta\lambda\right)} \right\rceil, \tag{44}$$

and for any time step $\tau \geq \tau_2$, we have $\|\boldsymbol{W}_L(\tau)^\top \boldsymbol{W}_L(\tau) - \boldsymbol{W}_{L-1}(\tau)\boldsymbol{W}_{L-1}(\tau)^\top\|_2 \leq \epsilon_2$. Applying Corollary B.8 with $\xi_1 = \sqrt{2}\epsilon_1$, $\xi_2 = \epsilon_2$, and the upper bound $r$ yields the desired result. □

*Proof of Theorem 4.3.* The conclusion follows from Proposition B.9; specifically, by replacing $\epsilon_1^2$ with $\epsilon_1$ and evaluating the order of $\tau_1'$ and $\tau_2$ in the proposition. By the condition of $\lambda \leq \alpha$ in Proposition B.9 and the definition of $m_\lambda$, we can bound $m_\lambda$ as a constant order depending only on the initialization. Consequently, the two terms in the upper bound of RNC1 are each multiplied by scales independent of $\lambda$ and $\mu$; thus, we have $\mathrm{RNC1}(\tau) = O\left((\sqrt{\epsilon_1} + \sqrt{\epsilon_2})^2\right) = O\left(\epsilon_1 + \epsilon_2\right)$. Finally, we evaluate the order of the upper bounds on $\tau_1'$ and $\tau_2$. Since we have $\log(1 - x) \approx -x$ for small $x$, the desired order expression of time steps is obtained. □

## C  EXPERIMENTAL DETAILS

The experiments on grokking in the main text follow the classification setup introduced in Liu et al. (2023a), where an MLP is trained on the MNIST dataset. The MLP has hidden dimensions $[784, 200, 200, 200, 10]$ with ReLU activations, and does not include normalization or dropout layers. The initialization scale is enlarged by a factor of eight across the entire network, and training is performed with the AdamW optimizer at a learning rate $1\mathrm{e}{-3}$. For Figure 2, we set the weight decay to $0.01$, while Figure 3 shows the results across multiple values of weight decay. The training set size is 1000, the batch size is 100, and the model is trained for 300,000 iterations. Compared to Liu et al. (2023a), we increased the number of layers in the MLP from three to four to examine behaviors in deeper architectures. As a consequence, we observe a slight instability in training accuracy just before fitting the training set in Figure 3. Nevertheless, the grokking behavior still clearly occurs.

For our experiments on the IB dynamics, we adopted two estimation methods for information-theoretic quantities: (i) MI estimation based on autoencoders and (ii) information estimation based on HSIC. As we describe below, each of these methods has been widely used in the literature, but they exhibit different characteristics. By employing multiple estimation methods, we aim to enhance the plausibility of the experimental results presented in our work.

**MI Estimation via Autoencoder.**  MI estimation via kernel density estimation (KDE) is widely used in IB studies, but it suffers from poor sample complexity in high-dimensional settings. To address this issue, we adopted the compression-based approach proposed by Butakov et al. (2024b), which performs MI estimation in a lower-dimensional space by applying dimensionality reduction via autoencoders.

Following Butakov et al. (2024b), we first train autoencoders on the input $X$ and the representation $Z$ and estimate differential entropies in low-dimensional latent spaces with dimensions $d_X = 4$ and $d_Z = 4$, respectively. The estimation is performed using the Kozachenko-Leonenko estimator (Kozachenko, 1987), which is based on the density estimation via $k$-nearest neighbors (k-NN). In practice, we follow Butakov et al. (2024b) and use a weighted variant of the estimator, as developed by Berrett et al. (2019). Although training a new autoencoder for each latent representation can be computationally demanding, Butakov et al. (2024b) showed that using a linear autoencoder, that is, principal component analysis (PCA), for compressing $Z$ yields competitive results. Therefore, we use PCA for compressing $Z$ in our experiments. For compressing $X$, we apply a toy CNN autoencoder in which both the encoder and decoder consist of five layers each.

**HSIC Estimation.**  Even approaches based on autoencoders require assumptions such as invertibility for accurate estimation and demand additional computational resources. To further support the findings of our study, we also conducted experiments based on normalized HSIC. Fukumizu et al. (2007, Theorem 4) shows that a normalized version of HSIC (to be defined below) coincides with the chi-square divergence between the joint distribution $P_{X,Y}$ and the product of marginals $P_X P_Y$.

Given that the MI is defined using the KL divergence instead of chi-square divergence, this suggests that the HSIC-based quantity can be interpreted as a variant of MI. More specifically, Gibbs & Su (2002, Theorem 5) shows that the chi-square divergence upper-bounds the KL divergence. Using HSIC as an information-theoretic quantity is a common practice in the literature on IB and learning algorithms (Ma et al., 2020; Pogodin & Latham, 2020; Wang et al., 2021; Jian et al., 2022; Guo et al., 2023; Wang et al., 2023; Sakamoto & Sato, 2024).

We now describe the estimation procedure along with the necessary preliminaries. The HSIC is a measure of statistical dependence between two random variables that can capture non-linear relationships. Suppose that we have two random variables $X$ and $Y$ on probability spaces $(\mathcal{X}, \mathcal{B}_X, P_X)$ and $(\mathcal{Y}, \mathcal{B}_Y, P_Y)$, respectively, where $\mathcal{B}_X$ and $\mathcal{B}_Y$ are the Borel $\sigma$-algebras on $\mathcal{X}$ and $\mathcal{Y}$. We consider functions that map elements of each sample space to real values. Let $\mathcal{F}$ and $\mathcal{G}$ be reproducing kernel Hilbert spaces (RKHS) with corresponding kernels $\kappa_{\mathcal{F}} : \mathcal{X} \times \mathcal{X} \to \mathbb{R}$ and $\kappa_{\mathcal{G}} : \mathcal{Y} \times \mathcal{Y} \to \mathbb{R}$. The mean $\mu_X$ is defined as an element of $\mathcal{F}$ such that $\langle \mu_X, f \rangle_{\mathcal{F}} = \mathbb{E}_X[f(X)]$ for all $f \in \mathcal{F}$. Similarly, let $\mu_Y$ and $\mu_{XY}$ denote the mean elements of $\mathcal{G}$ and $\mathcal{F} \otimes \mathcal{G}$, respectively. The cross-covariance operator $C_{XY} : \mathcal{G} \to \mathcal{F}$ is defined as a linear operator such that $\mathcal{C}_{XY} := \mu_{XY} - \mu_X \mu_Y$. Please note that $\mu_{XY} - \mu_X \mu_Y$ is an element of $\mathcal{F} \otimes \mathcal{G}$, but we can regard it as a linear operator from $\mathcal{G}$ to $\mathcal{F}$ by defining $\langle f, \mathcal{C}_{XY} g \rangle_{\mathcal{F}} = \langle fg, \mu_{XY} - \mu_X \mu_Y \rangle_{\mathcal{F} \otimes \mathcal{G}}$ for any $f \in \mathcal{F}$ and $g \in \mathcal{G}$. The Hilbert Schmidt norm of the linear operator $\mathcal{C} : \mathcal{G} \to \mathcal{F}$ is defined as $\|\mathcal{C}\|_{HS}^2 := \sum_{i,j} \langle \phi_i, \mathcal{C}\psi_j \rangle_{\mathcal{F}}^2$, where $\{\phi_i\}$ and $\{\psi_j\}$ are orthonormal bases of $\mathcal{F}$ and $\mathcal{G}$, respectively. We define the HSIC as $\mathrm{HSIC}(X, Y) := \|\mathcal{C}_{XY}\|_{HS}^2$, which is calculated using kernel functions as follows:

$$\begin{aligned}
\mathrm{HSIC}(X, Y) = {} & \mathbb{E}_{X,Y,X',Y'} \left[ \kappa_{\mathcal{F}}(X, X')\kappa_{\mathcal{G}}(Y, Y') \right] \\
& - 2\mathbb{E}_{X,Y} \left[ \mathbb{E}_{X'} \left[ \kappa_{\mathcal{F}}(X, X') \mid X \right] \mathbb{E}_{Y'} \left[ \kappa_{\mathcal{G}}(Y, Y') \mid Y \right] \right] \quad (45) \\
& + \mathbb{E}_{X,X'} \left[ \kappa_{\mathcal{F}}(X, X') \right] \mathbb{E}_{Y,Y'} \left[ \kappa_{\mathcal{G}}(Y, Y') \right],
\end{aligned}$$

where $(X', Y')$ is independent copy of $(X, Y)$. Given a dataset $\{(x_i, y_i)\}_{i=1}^N$ following $P_{X,Y}$, we can estimate the HSIC as $\mathrm{Tr}(\boldsymbol{K}_X \boldsymbol{H} \boldsymbol{K}_Y \boldsymbol{H})/(N-1)^2$, where we denote $\boldsymbol{K}_X, \boldsymbol{K}_Y, \boldsymbol{H} \in \mathbb{R}^{N \times N}$, $\boldsymbol{K}_{X,i,j} = \kappa_{\mathcal{F}}(x_i, x_j)$, $\boldsymbol{K}_{Y,i,j} = \kappa_{\mathcal{G}}(y_i, y_j)$, and the centering matrix $\boldsymbol{H} = \boldsymbol{I}_N - \boldsymbol{1}_N \boldsymbol{1}_N^\top / N$. In our experiments, motivated by the connection to the chi-square divergence described above, we use a normalized version of HSIC (nHSIC). Specifically, the nHSIC is defined as the Hilbert-Schmidt norm of the normalized cross-covariance operator, which is given by $\mathrm{nHSIC}(X, Y) := \|\mathcal{C}_{XX}^{-1/2} \mathcal{C}_{XY} \mathcal{C}_{YY}^{-1/2}\|_{HS}^2$. As in previous studies employing nHSIC for DNN analysis, we define the estimator as $\mathrm{Tr}[\boldsymbol{K}_X \boldsymbol{H} (\boldsymbol{K}_X \boldsymbol{H} + \epsilon N \boldsymbol{I}_N)^{-1} \boldsymbol{K}_Y \boldsymbol{H} (\boldsymbol{K}_Y \boldsymbol{H} + \epsilon N \boldsymbol{I}_N)^{-1}]$ and set $\epsilon = 1\mathrm{e}{-5}$.

# D    ADDITIONAL EXPERIMENTS

## D.1    GROKKING

**Decision Boundaries for Additional Class Pairs.**    Figure 2 showed how representations evolve during training on MNIST, but due to space limitations in the main text, we presented only the decision boundary between class $0$ and $1$. To demonstrate that the same phenomenon occurs for other class pairs as well, Figure 5 shows results for two additional pairs: class 4 vs. 5 and class 8 vs. 9. For these pairs, we observe the same trend: during the overfitting phase, training examples are already almost separable but still exhibit large within-class variance. As training proceeds, the training examples become more separated and more tightly clustered. Consequently, test examples are better separated and more concentrated in representation space, leading to improved generalization.

**Fashion-MNIST.**    We conduct experiments on Fashion-MNIST (Xiao et al., 2017) as additional experiments on a different dataset. Following the main text, we use a four-layer MLP with the same training configurations. Figure 6 shows the results, indicating that grokking also occurs on Fashion-MNIST. What is important here is not merely the occurrence of grokking, but rather that the decrease in the RNC1 score coincides with the increase in test accuracy, and that stronger weight decay accelerates this timing. These observations reinforce our theoretical results. As supplementary information, we also provide results when increasing the training set size from 1000 to 3000. In this case, the overall trend remains unchanged, but the test accuracy increases slightly earlier.

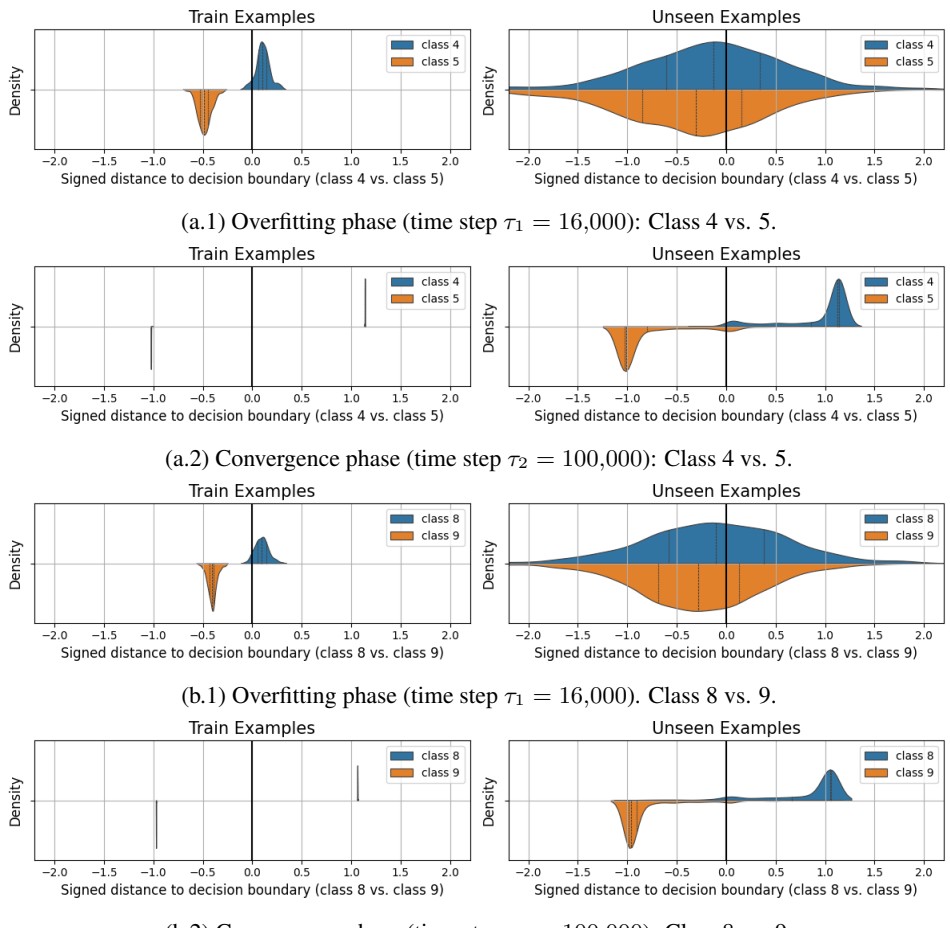

(a.1) Overfitting phase (time step $\tau_1 = 16{,}000$): Class 4 vs. 5.

(a.2) Convergence phase (time step $\tau_2 = 100{,}000$): Class 4 vs. 5.

(b.1) Overfitting phase (time step $\tau_1 = 16{,}000$). Class 8 vs. 9.

(b.2) Convergence phase (time step $\tau_2 = 100{,}000$): Class 8 vs. 9.

Figure 5: Margins of individual examples at two time steps during grokking. The model is a 4-layer MLP on the MNIST dataset. Here, we provide the corresponding plots for several class pairs other than the 0-1 pair shown in Figure 2.

**Modular Arithmetic.**   Previous studies on grokking have primarily focused on modular arithmetic tasks. For example, the addition task takes two integers $a$ and $b$ as input and outputs their sum modulo a prime number $p$, i.e., $(a + b) \mod p$. However, this setup is closer to a regression problem than a classification, as the outputs have an ordinal structure rather than being categorical labels. As our analysis is on classification, we consider this problem outside the scope of our study.

**CNN.**   In the main text, we examined grokking in a classification setting using MLP models, following Liu et al. (2023a). This choice is consistent with the theoretical analysis setting in Section 4, where neural collapse was studied with an MLP feature extractor. To further investigate whether grokking occurs in other architectures, we conducted additional experiments with CNNs. We trained a CNN consisting of two convolutional layers with max-pooling, followed by two fully connected layers. As in the MLP experiments, we attempted to scale the initialization of all layers by a factor of eight, but training was unstable. We therefore considered the modification scaling only the fully connected layers.

Figure 7 shows the results of changing the weight decay parameter $\lambda$. For all $\lambda$, test accuracy improves at the same time as training accuracy, and no grokking behavior is observed. Nevertheless, the results are consistent with our analysis: test accuracy improves in parallel with the progression of neural collapse. As for why there is no time lag between fitting the training set and the reduction of within-class variance, a possible explanation is that CNNs, due to their inductive bias of local invariance, already extract useful features during the fitting phase. Theorem 4.3 suggests that the

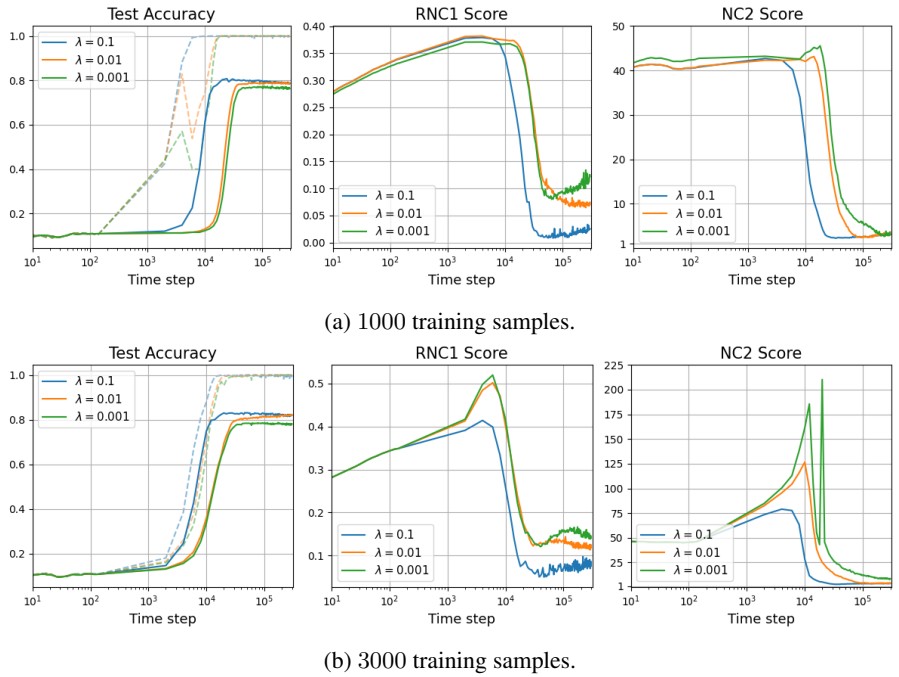

(a) 1000 training samples.

(b) 3000 training samples.

Figure 6: MLP trained on the Fashion-MNIST dataset with different weight decay coefficients $\lambda$. Test accuracy, RNC1, and NC2 scores are reported. In the test accuracy panel (left), the training accuracy is additionally shown in dashed lines of the same color to visualize grokking behavior. Results are averaged over five different seeds.

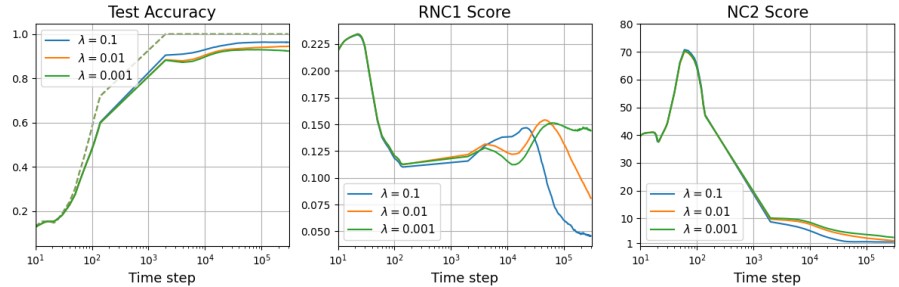

Figure 7: CNN trained on the MNIST dataset with different weight decay $\lambda$. Test accuracy, RNC1, and NC2 scores are reported. In the test accuracy panel (left), the training accuracy is additionally shown in dashed lines of the same color. Results are averaged over five different seeds.

fit of the output $f(\mathbf{X})$ to the labels $\mathbf{Y}$ is propagated to the preceding layer, i.e., the representation space, through the progression of weight balancedness. In contrast, if the representations are already well concentrated from the fitting phase, then no such delay arises after $f(\mathbf{X})$ fits the labels. This interpretation is supported by Figure 8, which shows the CNN results for the experiment in Figure 2 of the main text. Compared with the grokking scenario with MLPs in Figure 2, the CNN representations are already well collapsed at the point when the training set is first fitted (Figure 8a). Consequently, the margin distribution for test examples is relatively concentrated from the fitting phase, and as training progresses, its center becomes increasingly separated. Finally, Figure 7 also shows that when $\lambda = 0.1$, the RNC1 score continues to decrease after the training accuracy has reached $1.0$, and the test accuracy further improves. Thus, although this case is not grokking behavior, it demonstrates that continuing training after fitting the training set can further reduce the within-class variance and thereby improve test accuracy.

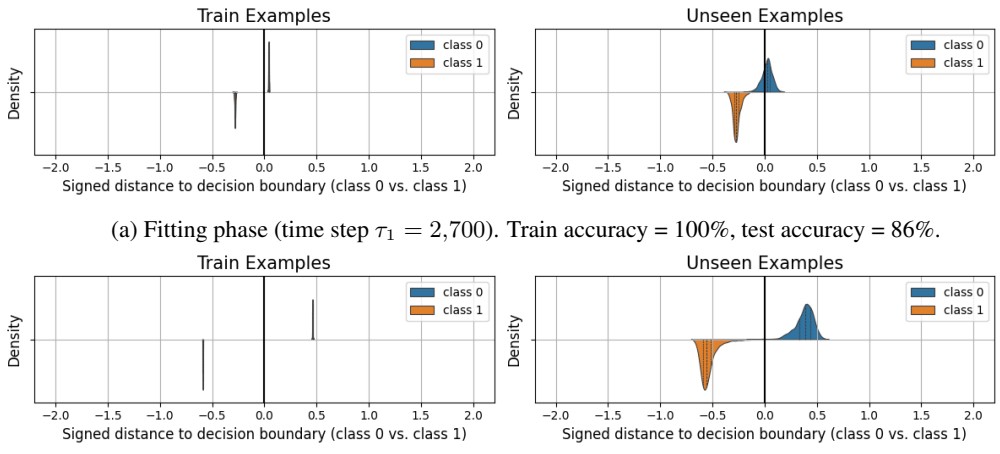

(a) Fitting phase (time step $\tau_1 = 2{,}700$). Train accuracy = 100%, test accuracy = 86%.

(b) Convergence phase (time step $\tau_2 = 100{,}000$). Train accuracy = 100%, test accuracy = 94%.

Figure 8: Margins at two time steps for a CNN trained on the MNIST dataset.

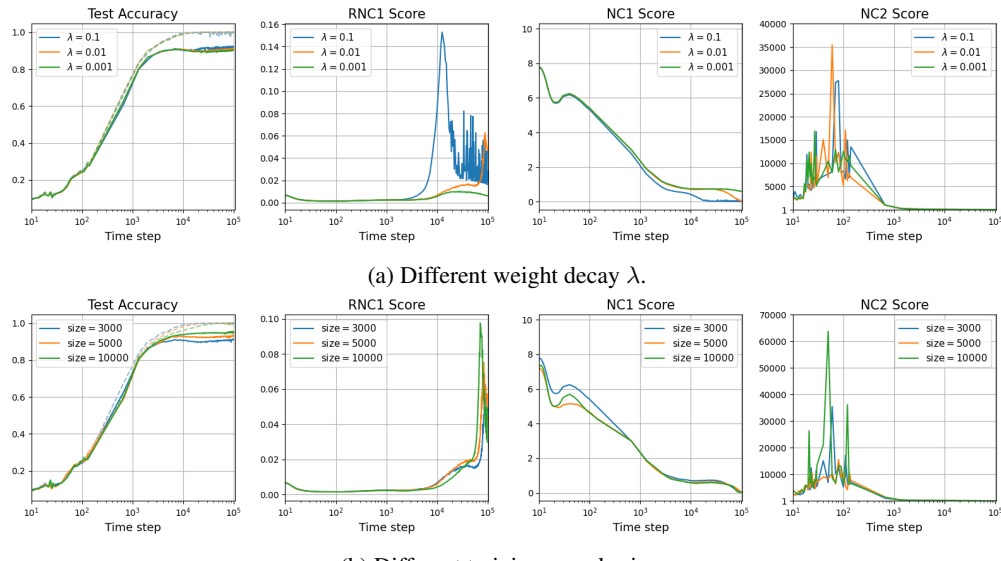

(a) Different weight decay $\lambda$.

(b) Different training sample sizes.

Figure 9: One-layer ViT trained on the MNIST dataset. Test accuracy, RNC1, NC1, and NC2 scores are reported. In the test accuracy panel (left), the training accuracy is additionally shown in dashed lines of the same color. Results are averaged over five different seeds.

**Transformer + MNIST.** We also conducted experiments with a transformer architecture (Vaswani et al., 2017) for MNIST classification. Specifically, we use a one-layer vision transformer(ViT) (Dosovitskiy et al., 2020) with hidden dimension $128$, four heads, and feedforward dimension $256$, without dropout. The input images are divided into patches of size $4$ and embedded with a convolutional layer, followed by learnable positional encoding and a class token. At the position of the class token, a linear head is attached for classification. Following prior work of grokking, we adopt an initialization scale of eight, but for training stability, the scaling is applied to the feedforward layers and the linear head.

Figure 9 shows the results when changing weight decay and training sample sizes, where no grokking behavior is observed. In this case, the RNC1 score and test accuracy show different trends, whereas the NC1 score decreases in accordance with the improvement in test accuracy. This reflects that, to account for test accuracy improvements, not only the reduction in the RNC1 score but also class mean separation must be considered. Theorem 3.2 captures both of these elements in

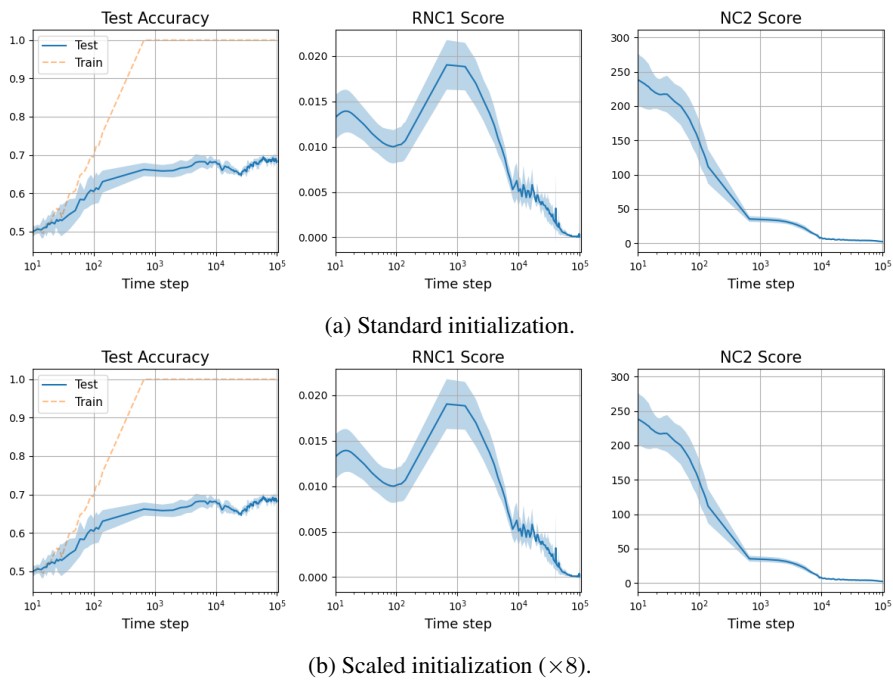

(a) Standard initialization.

(b) Scaled initialization ($\times 8$).

Figure 10: One-layer transformer encoder trained on SST-2. Results are averaged over five different seeds, and shaded areas correspond to one standard deviation.

the generalization bound. As shown in the grokking experiments in the main text, when the class means are separated, reducing the RNC1 score leads to improved test accuracy. In contrast, when the class mean separation is insufficient in the early stages of training, a small RNC1 score alone does not guarantee good classification performance. This aspect is reflected in the NC1 score; as discussed in Remark 4.2, unlike RNC1, the NC1 score incorporates the information on class-mean separation through its ratio with between-class variance. It explains why its decrease coincides with improvements in test accuracy in Figure 9.

**Transformer + Text Datasets.** To validate our findings across diverse datasets, we conducted experiments on three text classification benchmarks: **SST-2**, a binary sentiment classification task (Socher et al., 2013); **TREC-6**, a question classification task with six classes (Hovy et al., 2001; Li & Roth, 2002); and **AG-news**, a topic classification task with four news categories (Zhang et al., 2015). For preprocessing, we use the Hugging Face Bert WordPiece tokenizer (Devlin et al., 2019) solely for tokenization, restricting the vocabulary to tokens present in the training set. All embeddings, including unknown and padding tokens, are randomly initialized, and the maximum sequence length is set to $128$. The model configuration is the same as in the MNIST ViT experiment. Training is performed with a weight decay of $1e-2$ and a training set size of $3000$. We show results for both standard initialization and a variant where the initialization scale of the feedforward and linear layers is scaled up by a factor of $8$, building on the prior grokking studies.

The results are shown in Figure 10 (SST-2), Figure 11 (TREC-6), and Figure 12 (AG-news). In all cases, we observe little difference between different initialization scales, and grokking does not occur. As a consistent trend, the figures show that as the model fits the training set, the RNC1 score first peaks and then decreases. Notably, for SST-2 and AG-news, this decrease in RNC1 score coincides with a gradual improvement in test accuracy. While this behavior is not as abrupt as grokking, it supports our theoretical result that continuing training beyond the training accuracy plateau, driving further neural collapse, can benefit generalization.

## D.2 IB DYNAMICS

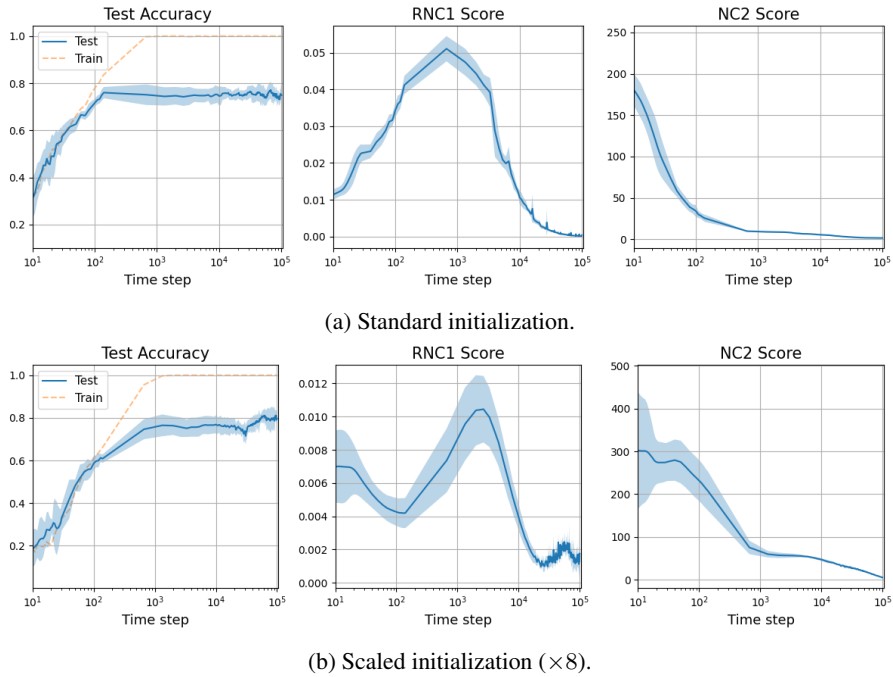

Figure 11: One-layer transformer encoder trained on TREC-6. Results are averaged over five different seeds, and shaded areas correspond to one standard deviation.

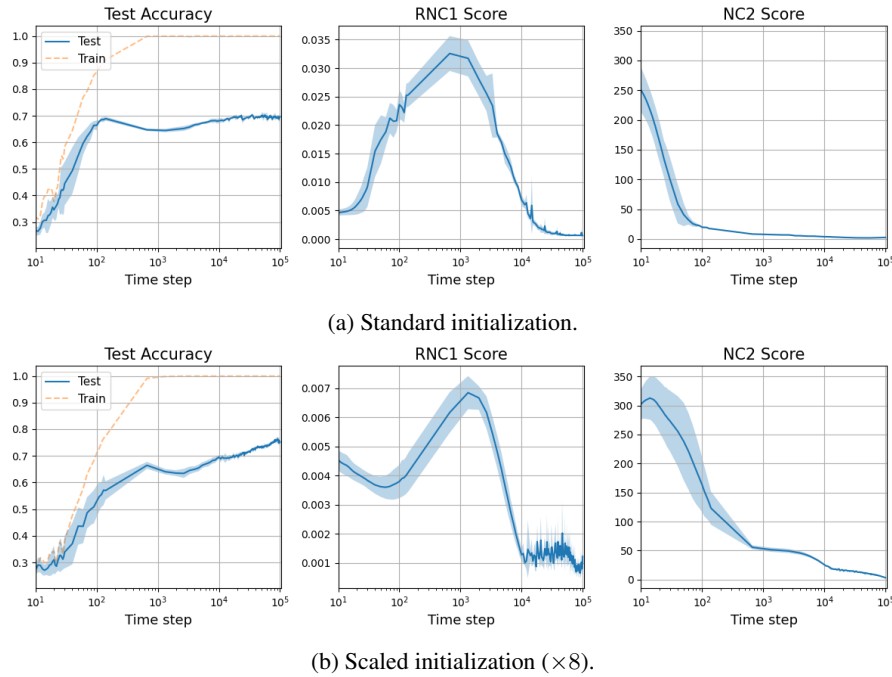

Figure 12: One-layer transformer encoder trained on AG-news. Results are averaged over five different seeds, and shaded areas correspond to one standard deviation.

**Fashion-MNIST.** As an additional experiment on a different dataset, we conducted experiments on Fashion-MNIST. The experimental setup is the same as that of Figure 6a, containing a four-layer MLP with scaled initialization. Figure 13 shows the results. As noted before, increasing the weight decay accelerates the decrease in the RNC1 score, and the same behavior is observed for

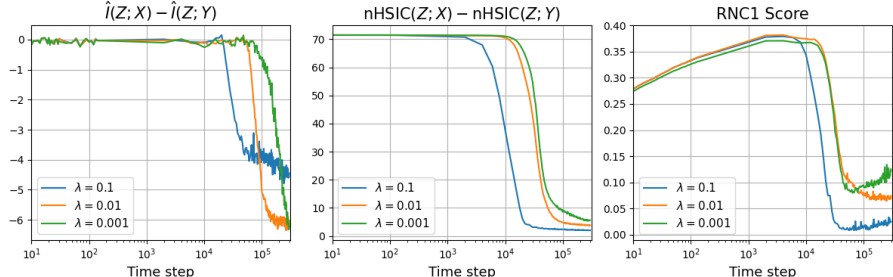

Figure 13: MLP trained on the Fashion-MNIST dataset with different weight decay coefficients $\lambda$. Dynamics of the redundant information (estimated via MI and nHSIC) and RNC1 scores are reported. Results are averaged over five different seeds.

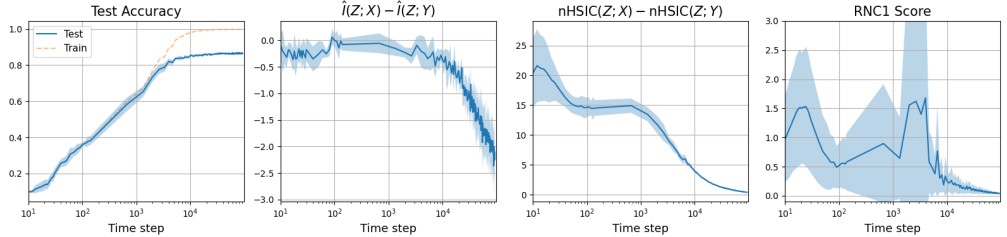

Figure 14: ResNet18 trained on the CIFAR10 dataset. Dynamics of test accuracy, redundant information (estimated via MI and nHSIC), and RNC1 scores are reported. In the test accuracy panel (left), the training accuracy is shown in dashed lines. Results are averaged over five different seeds, and shaded areas correspond to one standard deviation.

redundant information measured by MI and nHSIC. Similar to the MNIST experiments in Figure 4, the redundant information estimated via MI decreases slightly later than that estimated via nHSIC. In either case, the results indicate that the decrease in the RNC1 score leads to the reduction of these information measures.

**ResNet + CIFAR10.** We also conducted IB experiments in a more standard setting, training ResNet18 (He et al., 2016) on the CIFAR10 dataset (Krizhevsky, 2009). In the grokking experiments, it was important to delay the decrease of the RNC1 score relative to the fit to the training set, which was achieved by adopting a large initialization scale and a small sample size. In contrast, the reduction of redundant information in the IB principle does not necessarily require such a timing discrepancy. Therefore, in this experiment, we used the standard initialization scale and the full training set of 50,000 examples. Figure 14 shows the results, with test accuracy shown on the left for the reference. For both MI and nHSIC, redundant information decreases as training progresses. When compared with the behavior of the RNC1 score, the trends are similar particularly in the later phase of training, supporting our theoretical result that links the reduction of IB superfluous information to the decrease of the RNC1 score. The left panel shows that this later compression phase corresponds to the period after the training set has already been fit. This suggests that continuing training to promote neural collapse is beneficial not only for generalization but also from the perspective of the IB principle.

