# OpenReview forum: "Explaining Grokking and Information Bottleneck through Neural Collapse Emergence"
_ICLR.cc/2026/Conference — ICLR 2026 Poster_

### Official Review · Reviewer_oBWN · 2025-10-30

**Soundness:** 3
**Presentation:** 3
**Contribution:** 3
**Rating:** 4
**Confidence:** 3

**Summary:**

The paper investigates learning phaes of neural nets focusing on the later training stages using wihtin-class variance as a key investigative lense.

**Strengths:**

* Important problem: Theory is generally lacking in this area
* Good alignment of theory and practice.
* Well written.

**Weaknesses:**

* There are some concerns regarding the suitability of in-class variance, which appear critical   (see questions).
* As for any theory, there are a number of assumptions.  and lack of common architectural elements
* There are concerns regarding novelty: Some of the findings appear known, some approaches seem to overlap with prior work, some key statements appear to be known, though not in this context and under a different analytical lense -  but the paper still of interest despite potential overlaps, but clarification is needed in this regard (see questions).
* Lack of practical implications (though not a must for a theory paper)

(Note, the initial judgement is more on the negative side due to a number of concerns - in particular the in-class variance as metric, but given questions are clarifyable an increase in score is appropriate as this is overall a strong paper)

**Questions:**

* How is this behavior if you include different kinds of normalization?
* Could you intuitively state the impact of assumption A.5 (in appendix)? In what sense, does it restrict initialization (e.g. compared to He or other initialization methods)?

* A variance decrease is also possible by simple scaling, e.g., say X are inputs to a layer then we have var[WX] for some weight matrix W, now if we scale X, i.e., we have cX with c<1, we have c^2*var[WX].
If you use regularization on weights you naturally scale weights and push them towards 0. So from this angle the metric appears not very suitable.


* You state "An intriguing observation in the existing information plane work is that, in the late stage of training, DNNs tend to compress I(Z; X) while preserving I(Z; Y ), thereby moving toward a more optimal solution with respect to the IB objective in Equation (1).

This observation has already been made outside the IB framework, by looking at the reconstruction loss L(X,Z) over time (see [1] below). That is the paper argues that reconstructing the input X given intermediate representation Z gets better (L(X,Z) decreases) before increasing again (L(X,Z) increases), at the same time cross entropy CL(X,Y) loss only reduces, which is qualitatively identical to your Figure 1 showing I(Z,Y), I(Z,X), except that they also show a third phase, i.e., initially L(Z,X) remains constant.  Do you agree?

[1] Schneider, J., & Prabhushankar, M. (2024). Understanding and leveraging the learning phases of neural networks. In Proceedings of the Thirty-Eighth AAAI Conference on Artificial Intelligence and Thirty-Sixth Conference on Innovative Applications of Artificial Intelligence and Fourteenth Symposium on Educational Advances in Artificial Intelligence (pp. 14886-14893).


* Why should in-class variance be a good measure? In particular, you show that it decreases, but it a better measure might something that changes across the phases ? Won't it?

*  [2] (see below) argues that first the mean of the samples are fit, followed by those not well-aligned which might actually strongly impact representations though only minimally impacting the loss, which is fairly intuitive. Does not the initial convergence to the mean (per class) also apply to your formulation?   Essentially you minimize the population within class variance (Definition 3.1) if g(X) ~ Mean_(X,Y=c) g(X|c) ~ E_{X|Y=c}[g(X)].  In particular, the RNC1 score seems to measure just that and directly corresponds to the L2-norm used in [2]. Could you discuss the differences and similarities?
The paper [2] argues that after the majority has been fit, learning slows down as but there can still be substanital changes -- the paper says a few (strongly different samples from the mean) can be repsonsible for this using the dot product. This seems to be very different from your approach as you are only concerned with the majority, are you? Or is it also relevant in your case ?

[2] Learning in NN: from fitting most to fitting a few. Neural Computing and Applications, 37(28), 23423-23446.

---

> ### Author Response · Authors · 2025-11-18
> **Reply to Reviewer oBWN (1/4)**
>
> We appreciate your extensive and detailed feedback on our work. We have addressed each of your concerns and revised our draft. We hope that our responses below adequately resolve your concerns and will positively impact your evaluation.
>
> ---
> > (Q1) How is this behavior if you include different kinds of normalization?
>
> Since your question may refer to several different components, we address each possibility below.
> 1. Weight decay vs. other forms of regularization.
>    Our work focuses on weight decay, which is among the most widely used forms of regularization (e.g., AdamW in LLM training). Several prior works explicitly analyzed grokking under weight decay [3, 4, 5], and our contribution follows this established line of work.
>    On the other hand, there are reports of grokking without weight decay in different problem settings [6]. In such cases, implicit reguralization, such as the bias to the max-margin solutions, may potentially play a similar role. Exploring this possibility is an interesting direction for future work, but is outside the scope of the present paper.
> 2. Normalization layers.
>    We do not study normalization layers such as LayerNorm and BatchNorm in this paper. However, all arguments up to Section 3 apply to a general feature extractor $g$, and Theorem 4.1 continues to hold after adjusting for output scale. Thus, our high-level conclusions remain valid even when normalization layers are included. A precise analysis of how normalization layers affect the training dynamics in Theorem 4.3 would require further analysis, and we have added this point as a future research direction.
>
> [3] Liu, Ziming, et al. "Towards understanding grokking: An effective theory of representation learning." NeurIPS2022.
> [4] Liu, Ziming, Eric J. Michaud, and Max Tegmark. "Omnigrok: Grokking Beyond Algorithmic Data." ICLR2023.
> [5] Lyu, Kaifeng, et al. "Dichotomy of Early and Late Phase Implicit Biases Can Provably Induce Grokking." ICLR2024.
> [6] Kumar, Tanishq, et al. "Grokking as the transition from lazy to rich training dynamics." ICLR2024.
>
> ---
> > (Q2) Could you intuitively state the impact of assumption A.5 (in appendix)? In what sense, does it restrict initialization (e.g. compared to He or other initialization methods)?
>
> Intuitively, Assumption A.5 requires that the contribution of the second layer at initialization remains small relative to other layers. This condition can be ensured by increasing the initialization scale of all layers except the second layer (which increases the left-hand side), and by making the second-layer scale sufficiently small (which upper-bounds the initial training-loss term on the right-hand side).
> Furthermore, by relaxing Assumption A.3 from $d_1 \geq N$ to $d_1 = \Omega(N)$, **Assumption A.5 is also satisfied under the standard He/LeCun initialization scale**. This follows from the argument in Appendix C of [7].
>
> Assumption A.3-A.5 are exactly the assumptions adopted in the prior work [8] to analyze the training-loss dynamics of DNNs and establish a lower bound on the Jacobian. Our paper follows these assumptions; **we introduce no new assumptions**. Moreover, these assumptions are required only for Theorem 4.3, which characterizes the time scale of the training-set fitting. All our results up to Theorem 4.1, including those relating grokking, IB, and neural collapse, do **not** rely on these assumptions.
> To clarify this point, **we have added a new remark titled "Remark A.6: (Validity of the Assumptions)"** immediately after Assumptions A.3-A.5, where we discuss their necessity and their trade-offs with other commonly used assumptions in the literature.
>
> [7] Nguyen, Quynh N., and Marco Mondelli. "Global convergence of deep networks with one wide layer followed by pyramidal topology." NeurIPS2020.
> [8] Arthur Jacot, Peter S ́uken ́ık, Zihan Wang, and Marco Mondelli. Wide neural networks trained with weight decay provably exhibit neural collapse. ICLR2025 Oral.

---

> ### Author Response · Authors · 2025-11-18
> **Reply to Reviewer oBWN (2/4)**
>
> ---
> > (Q3) A variance decrease is also possible by simple scaling, e.g., say $X$ are inputs to a layer then we have $\mathrm{Var}(WX)$ for some weight matrix W, now if we scale X, i.e., we have $cX$ with $c<1$, we have $c^2* \mathrm{Var}(WX)$. If you use regularization on weights you naturally scale weights and push them towards 0. So from this angle the metric appears not very suitable.
>
> **All of our analysis and experiments are based on the scale-invariant within-class variance** defined in Definition 3.1, not on $\mathrm{Var}(WX)$.
> You are absolutely right that the naive variance $\mathrm{Var}(WX)$ is scale-sensitive: if the weights shrink due to regularization, then $\mathrm{Var}(WX)$ would decrease trivially. However, this issue does not arise here because the scale-invariant variance in Definition 3.1 cancels out any global scaling of the representation. Specifically, even if the network representation scales as $Z = g(Z) \mapsto cg(Z)$ for $c < 1$, the quantity in Definition 3.1 is invariant to such scaling because the denominator $B_g$ cancels the multiplicative factor. Therefore, the variance we study does **not** decrease simply because the weights shrink; it only decreases when the relative geometry of the representations becomes more concentrated.
>
> Your question highlights an important aspect of our approach: by using a scale-invariant notion of within-class variance, the analysis cleanly separates changes in output scale from genuine geometric contraction of class variance. To emphasize this point, **we have added the following clarification before Definition 3.1**:
> *In particular, the scale-invariant formulation in Definition 3.1 ensures that this quantity reflects genuine geometric concentration, disentangled from changes in output scale.*
>
> ---
> > (Q4) This observation has already been made outside the IB framework, by looking at the reconstruction loss L(X,Z) over time (see [1] below). That is the paper argues that reconstructing the input X given intermediate representation Z gets better (L(X,Z) decreases) before increasing again (L(X,Z) increases), at the same time cross entropy CL(X,Y) loss only reduces, which is qualitatively identical to your Figure 1 showing I(Z,Y), I(Z,X), except that they also show a third phase, i.e., initially L(Z,X) remains constant. Do you agree?
> > [1] Schneider, J., & Prabhushankar, M. (2024). Understanding and leveraging the learning phases of neural networks.
>
> Thank you very much for pointing us this interesting work. Although their analysis is based on reconstruction loss rather than mutual information, we agree that the overall high-level scenario of "late-stage information compression" is consistent with the IB intuition. **We have now cited this paper in the related work section**.
> Below we discuss the differences in detail.
>
> Using the reconstruction loss $L(X, Z)$ as a proxy or variational lower bound for $I(X; Z)$ has been commonly done in mutual-information-based representation learning [9, 10, 11]. However, reconstruction loss fundamentally appears as a lower bound on mutual information. This implies a strict limitation:
> - A **decrease** in reconstruction loss indicates that $I(X; Z)$ is **not decreasing** (i.e., relevant information is preserved),
> - but an **increase** in reconstruction loss does **not** imply that $I(X; Z)$ is decreasing, as the lower bound provides no information about actual compression.
>
> This is precisely why in our paper, Proposition 3.3 uses loss decrease to argue that relevant information is preserved, but Theorem 3.4, which proves redundant-information compression, must rely on an entirely different argument, not a reconstruction-loss bound.
> Therefore, **while reconstruction-loss behavior can serve as a proxy for IB dynamics, it cannot explain IB dynamics**. In other words, although both approaches identify a late-phase reduction of redundant information, the reconstruction-loss view does not provide a direct characterization of the mutual information dynamics.
>
> Interestingly, Theorem 3.4 in our paper shows that redundant information decreases via the reduction of within-class variance. Since the mutual information satisfies the relation mentioned above: $I(X; Z) \geq -L(X;Z) + const$, our result can be interpreted as providing a theoretical explanation for the observed increase in reconstruction loss in [1]. This connection is indeed insightful.
> Finally, we note that our results characterize the compression dynamics through within-class variance, and that our theoretical results hold for general DNNs without relying on simplified data models or equal-class-size assumptions.
>
> [9] Hjelm, R. Devon, et al. "Learning deep representations by mutual information estimation and maximization." ICLR2019.
> [10] Poole, Ben, et al. "On variational bounds of mutual information." ICML2019.
> [11] Wang, Yulin, et al. "Revisiting locally supervised learning: an alternative to end-to-end training." ICLR2021.

---

> ### Author Response · Authors · 2025-11-18
> **Reply to Reviewer oBWN (3/4)**
>
> ---
> > (Q5) Why should in-class variance be a good measure? In particular, you show that it decreases, but it a better measure might something that changes across the phases? Won't it?
>
> We would like to clarify that we do **not** introduce (rescaled) within-class variance as an ad-hoc metric for explaining phase transitions. Rather, **this quantity arises naturally when analyzing the second phase of grokking, and the compression phase in the IB dynamics**. Indeed, within-class variance appears directly in Theorem 3.2 and 3.4, which characterize these phenomena. In contrast, the first phase, namely training loss converge in grokking (Theorem 4.3) or relevant-information preservation in IB (Proposition 3.3), is governed by entirely different mechanisms and does **not** involve within-class variance.
>
> Your question suggests that the metric might decrease monotonically across phases. However, we do not claim monotonic behavior. Theorem 4.3 only states that RNC1 score becomes sufficiently small eventually and that reaching this regime requires a long time scale. The theorem makes no claim about intermediate behavior; indeed, the proof explicitly assumes completion of training-set interpolation (the end of phase 1) before establishing the contraction of the RNC1 score. Consistent with this, Figure 3 shows that within-class variance may increase before decreasing, phase-transition behavior.
>
> In summary, our framework clearly separates the mechanisms governing each phase and explains their difference in timing through Theorem 4.3, which establishes that these phases operate on distinct time scales. To highlight that (rescaled) within-class variance arises naturally from the theoretical analysis, and is not an ad-hoc choice, **we have added the following clarification before Definition 3.1**:
> *We remark that this metric is not introduced as an ad-hoc measure; rather, it naturally arises from our analysis of the second-phase dynamics presented below.*
>
> ---
> > (Q6) [2] (see below) argues that first the mean of the samples are fit, followed by those not well-aligned which might actually strongly impact representations though only minimally impacting the loss, which is fairly intuitive. Does not the initial convergence to the mean (per class) also apply to your formulation? Essentially you minimize the population within class variance (Definition 3.1) if $g(X) \sim Mean_{(X,Y=c)}[g(X|c)] \sim E_{X|Y=c}[g(X)]$. In particular, the RNC1 score seems to measure just that and directly corresponds to the L2-norm used in [2]. Could you discuss the differences and similarities? The paper [2] argues that after the majority has been fit, learning slows down as but there can still be substanital changes -- the paper says a few (strongly different samples from the mean) can be repsonsible for this using the dot product. This seems to be very different from your approach as you are only concerned with the majority, are you? Or is it also relevant in your case ?
> > [2] Learning in NN: from fitting most to fitting a few. Neural Computing and Applications, 37(28), 23423-23446.
>
> Thank you very much for pointing us to this interesting work. Similar to our paper, [2] analyzes two-phase training dynamics, and it characterizes the learning phases through reconstruction loss, like the paper mentioned in Q4. The novelty of [2] lies in further decomposing the fitting phase into prototypical versus atypical samples, arguing that the latter can still significantly affect representations despite contributing little to the loss. This perspective based on sample-level geometry is highly insightful. Below we address your questions one by one.
>
> > (Q6.1) Does not the initial convergence to the mean (per class) also apply to your formulation?
>
> While our paper also studies a two-phase training dynamics, the objects of analysis differ from [2]. The two phases in [2] correspond to (1) fitting prototypical samples (close to class means), followed by (2) fitting atypical / noisy samples that affect representations. In contrast, in our frame work **the entire fitting of training set constitutes the first phase**, and the second phase refers to a distinct mechanism: the reduction of redundant information arising from the progression of neural collapse. Therefore, although both frameworks have a "two-phase" structure, the boundaries and underlying mechanisms of the phases are fundamentally different.
>
> > (Q6.2) Could you discuss the differences and similarities?
>
> The L2-norm quantity used in [2] is conceptually very different from our RNC1 score. In [2], atypicality is defined by geometric deviation in the **input space**, whereas in our paper, $g$ is a learned feature extractor, and RNC1 measures (rescaled) within-class variance in the **learned representation space**. Even samples that are atypical in the input space can be mapped by a learned feature extractor $g$ to align with the class mean vector, as predicted by neural collapse.

---

> ### Author Response · Authors · 2025-11-18
> **Reply to Reviewer oBWN (4/4)**
>
> > (Q6.3) This seems to be very different from your approach as you are only concerned with the majority, are you? Or is it also relevant in your case ?
>
> We do **not** focus only on the majority. Minority samples contribute to the within-class variance through their representations $g(x)$ and therefore affect both the population within-class variance and RNC1 score.
> As discussed above, [2] further decomposes our "first fitting phase" into sub-phases involving prototypical and atypical samples. This constitutes a finer-grained layering of the learning dynamics. Our analysis is compatible with this view rather than contradictory. Thus, our results can be combined with insights from [2] to offer an even more detailed view of late-phase dynamics.
>
> We believe our answers above sufficiently clarify the differences from [2]. Finally, **we now cite [2] (together with [1]) in the related work section** as an example of phase-based learning analysis via reconstruction loss.
>
>
> ---
> Based on our responses above, we believe that we have sufficiently addressed the reviewer’s stated weaknesses:
>
> > - There are some concerns regarding the suitability of in-class variance, which appear critical (see questions).
> > - As for any theory, there are a number of assumptions. and lack of common architectural elements
> > - There are concerns regarding novelty: Some of the findings appear known, some approaches seem to overlap with prior work, some key statements appear to be known, though not in this context and under a different analytical lense - but the paper still of interest despite potential overlaps, but clarification is needed in this regard (see questions).
>
> We are confident that our clarifications directly resolve these points while also reinforcing the novelty of our contribution. Finally, we additionally provide a response regarding the practical implications, as requested.
> > (Q7) Lack of practical implications (though not a must for a theory paper)
>
> Regarding practical implications, our work on late-phase phenomena such as grokking and IB dynamics provides useful useful guidance for real-world training method.
> **First**, our result suggest that monitoring quantities such as the rescaled within-class variance in our work can provide a indicator of whether a model is approaching the beneficial late training phase. For example, grokking requires continuing training even after the model perfectly fits the training set, which means that techniques such as early stopping would prematurely end training before reaching the desirable solution. In practice, it is unclear how long one should continue training, or even whether grokking (or other late-phase improvements) will eventually emerge at all. This uncertainty makes it difficult to allocate computational resources and training time appropriately. Our framework provides a representation-based signal that can guide this decision.
> **Second**, our analysis highlights the importance of weight-decay regularization in accelerating this transition. While there is a trade-off with model expressivity, our findings suggest that an appropriate degree of weight decay may enable reaching the desirable solution significantly more quickly.
>
> **We have added the following sentence summarizing these practical implications** into the Conclusion of the revised manuscript.
> *Beyond the theoretical perspective, our results also provide practical implications: tracking quantities such as the rescaled within-class variance can help determine when continued training will be beneficial, and weight decay can accelerate the transition to this late-phase regime.*

---

> > ### Comment · Reviewer_oBWN · 2025-11-22
> > **Thank you, good response, few details remaining, score increased.**
> >
> > Thanks, the response clarified major concerns and I increased the score.
> >
> > You write that: ... an increase in reconstruction loss does not imply that I is decreasing, as the lower bound provides no information about actual compression...
> >
> > Let assume a perfect decoder than:
> >
> > E[recon loss]=H(X∣Z), I(X;Z)=H(X)−H(X∣Z)=H(X)−E[recon loss].
> >
> > So, it is not just a lower bound but rather you can use reconstruction loss instead of I(X,Z). Do you agree?

---

> ### Author Response · Authors · 2025-11-24
>
> We truly appreciate the additional follow-up and your positive update to the score.
>
> You are absolutely right that under a perfect decoder, the equality you wrote indeed holds. In that idealized case, one can track the behavior of mutual information directly through the reconstruction loss.
> However, in practice, a neural network decoder is far from perfect due to limited expressivity and the fact that it is trained only on the finite training set.
>
> If you have any further concerns or suggestions for strengthening the paper, we would be very happy to address and incorporate them.

---

### Official Review · Reviewer_ud5o · 2025-10-31

**Soundness:** 2
**Presentation:** 2
**Contribution:** 2
**Rating:** 4
**Confidence:** 3

**Summary:**

This paper attempts to provide geometric formulation for grokking’s delayed generalization. They do so by showing that grokking is the combination of a deep neural network following IB dynamics and achieving within-class variance collapse. It derives bounds linking this variance collapse to both generalization error and redundant mutual information, and shows empirically that variance collapse (RNC1) lags training-loss convergence under weight decay. The result is a heuristic unification of IB, grokking, and NC1.

**Strengths:**

This paper reframes grokking and IB as natural outcomes of a slowly emerging regularity principle (neural collapse). The paper is well-written, offers clear theorems that formalize variance–generalization connections, and provides empirical breadth.
The time-scale separation between fitting and collapse is a genuine insight for grokking dynamics, and the presentation is careful and reproducible.

**Weaknesses:**

However, the analysis treats neural collapse only through NC1, uses an one-sided IB inequality (Thm 3.4), and replaces Papyan 2020’s ETF-based NC2 metric with a condition-number surrogate that cannot ensure ETF geometry. The paper’s main claim is a unification of grokking and information bottleneck (IB) dynamics through neural collapse (NC). This is conceptually interesting but not rigorously established.

This unification is heuristic not proven: The entire IB connection rests on a one-sided variance-based upper bound (Thm 3.4) that does not prevent degenerate or trivial solutions (when compression is more important than capturing any relevant information).

Theoretical results rely on strong simplifications (smooth activations, squared loss, pyramidal architecture, Assumption A.3, A.4, A.5). The theoretical guarantees in Theorem 4.3 depend on highly restrictive conditions (pyramidal architecture, smooth activations, squared-loss training) that differ from the experimental setups (ReLU + cross-entropy) . Theorem 4.3 and its proof explicitly assume optimization of the squared loss with weight decay, as given in Section 4.2.  All the convergence results refer to that MSE-based objective, not to the cross-entropy loss used in the experiments.

I also think replacing the NC2 metric with a condition-number surrogate without justification is sloppy. The co-occurrence of NC1 and NC2 is critical for the normative statement of neural collapse to hold (theorem 3 in Papyan 2020). NC2 is strictly defined as equal angles and equal norms both satisfied. While when condition number = 1, the perfectly isotropic mean configuration coincides with ETF. Many non-ETF configurations can also yield small condition numbers.

**Questions:**

1) Could flatness or implicit regularization, rather than NC1 dynamics, explain the delayed generalization you observe?

2) Is thm 3.4 a tight bound or a loose ceiling?

---

> ### Author Response · Authors · 2025-11-18
> **Reply to Reviewer ud5o (1/4)**
>
> We appreciate your extensive and detailed feedback on our work. We have addressed each of your concerns and incorporated the corresponding clarifications into the revised version. We hope that our responses below adequately resolve your concerns and will positively impact your assessment.
>
> ---
> > (Q1) The analysis treats neural collapse only through NC1. The paper’s main claim is a unification of grokking and information bottleneck (IB) dynamics through neural collapse (NC). This is conceptually interesting but not rigorously established.
>
> Neural collapse in our paper acts as a **sufficient condition** for establishing the connection between grokking and IB; our analysis does not require the full NC1-NC4 structure.
> We agree that NC1 represents only one aspect of neural collapse as defined in [Papyan+, 2020]. However, **our goal is not to prove the neural collapse itself**. Rather, our aim is to start from neural collapse and extend the discussion to grokking and IB dynamics. Therefore, assuming all four components of neural collapse would not affect any of our theoretical results, and relying solely on NC1 actually makes our claims more general, as they require fewer assumptions. This directional relationship is reflected both in the title and in Figure 1. Moreover, in Theorem 4.3, **we go beyond simply assuming NC1 and explicitly analyze the training dynamics leading to NC1** under gradient-based optimization.
>
> We address the question about NC2 and its condition-number surrogate in our response to Q4.
>
> ---
> > (Q2) The entire IB connection rests on a one-sided variance-based upper bound (Thm 3.4) that does not prevent degenerate or trivial solutions (when compression is more important than capturing any relevant information).
>
> The IB connection in our analysis is **not** limited to Theorem 3.4, and **we have already addressed the preservation of relevant information in Proposition 3.3**.
> It is true that Theorem 3.4 captures only the compression side of the IB framework. In contrast, Proposition 3.3 shows the training progresses in a direction that preserves relevant information. Considering these two results together enables us to characterize the IB trade-off. Your concern is partially valid as Theorem 3.4 alone would indeed permit degenerate solutions, for example, collapse to single representation. However, such degenerate cases are ruled out by Proposition 3.3, which ensures the relevant information cannot vanish during the optimization.
>
> To make this clear, **we have updated Figure 1 to more explicitly reflect this argument**. In particular, we annotate:
> - "Phase 1: Preservation (Proposition 3.3)" with larger font
> - "Phase 2: Compression (Theorem 3.4)" explicitly
>
> ---
> > (Q3) Theoretical results rely on strong simplifications (smooth activations, squared loss, pyramidal architecture, Assumption A.3, A.4, A.5). The theoretical guarantees in Theorem 4.3 depend on highly restrictive conditions (pyramidal architecture, smooth activations, squared-loss training) that differ from the experimental setups (ReLU + cross-entropy) . Theorem 4.3 and its proof explicitly assume optimization of the squared loss with weight decay, as given in Section 4.2. All the convergence results refer to that MSE-based objective, not to the cross-entropy loss used in the experiments.
>
> **The assumptions used in Theorem 4.3 are standard in the theoretical analysis of gradient-descent dynamics in DNNs, and our paper does not introduce any assumptions beyond what is assumed in prior work**. Importantly, these assumptions are used only in Theorem 4.3 to characterize the time scale of fitting the training set; the theoretical results unifying grokking, IB, and neural collapse up to Theorem 4.1 do not rely on these assumptions at all. Moreover, these assumptions are also adopted in recent analyses of gradient dynamics accepted at top‑tier venues, e.g., NeurIPS, ICML, ICLR, indicating that they are reasonable within the theoretical community.
> Below we detail how each assumption corresponds to prior work. (continued in the next reply)

---

> ### Author Response · Authors · 2025-11-18
> **Reply to Reviewer ud5o (2/4)**
>
> The derivation of neural collapse from gradient descent, instead of from the unconstrained feature model (UFM), was pioneered by [1], and Theorem 4.3 adopts exactly the same assumptions. Specifically, in order to analyze the convergence of the DNN training loss, one typically imposes conditions ensuring a positive lower bound on the Jacobian with respect to the parameters. There are multiple well-established ways to ensure this, for example, width-based assumptions across all layers [2, 3, 4] and other pyramidal topology + mild width requirement [5, 6, 7]. Our Assumptions A.3-A.5 can be replaced by any of these alternatives. For instance, [3, 4] analyze ReLU networks instead of smooth activations, but require all layers to be sufficiently wide. In contrast, the pyramidal topology assumption (Assumption A.3) removes the need for width assumptions on every layer and replaces them with a more realistic setting: a wide first layer followed by a narrowing architecture.
> The smooth activation assumption (Assumption A.4) is widely used in convergence analysis and appears in independent works such as [5, 6, 8, 9]. Smooth leaky ReLU satisfies this assumption and can approximate ReLU arbitrarily well for suitable choices of $\gamma$ and $\beta$. Assumption A.5 concerns initialization, and it can be satisfied by choosing the second-layer scale sufficiently small. Moreover, by relaxing Assumption A.3 from $d_1 \geq N$ to $d_1= \Omega(N)$, Assumption A.5 can be shown to hold under standard He/LeCun initialization, which follows directly from Appendix C of [5]. Finally, the use of squared loss is standard in this line of work: it is used both in the gradient-descent NTK-style analyses [1-5] and in the theoretical neural collapse literature [10-13].
> As also discussed in [1], once the training set is interpolated, the second part of the analysis (the time step $\tau_2$ in Theorem 4.3) that shows the emergence of neural collapse does not rely on these assumptions. We emphasize again that none of our results except Theorem 4.3 depend on Assumptions A.3-A.5.
>
> In response to your concern, **we have added a new remark titled "Remark A.6: (Validity of the Assumptions)"** immediately after Assumptions A.3-A.5, where we discuss in detail the necessity of these assumptions and provide the above clarifications in the revised draft.
>
>
> [1] Arthur Jacot, Peter S ́uken ́ık, Zihan Wang, and Marco Mondelli. Wide neural networks trained with weight decay provably exhibit neural collapse. ICLR2025 Oral.
> [2] Du, Simon, et al. "Gradient descent finds global minima of deep neural networks." ICML 2019.
> [3] Allen-Zhu, Zeyuan, Yuanzhi Li, and Zhao Song. "A convergence theory for deep learning via over-parameterization." ICML2019.
> [4] Zou, Difan, and Quanquan Gu. "An improved analysis of training over-parameterized deep neural networks." NeurIPS2019.
> [5] Nguyen, Quynh N., and Marco Mondelli. "Global convergence of deep networks with one wide layer followed by pyramidal topology." NeurIPS2020.
> [6] Bombari, Simone, Mohammad Hossein Amani, and Marco Mondelli. "Memorization and optimization in deep neural networks with minimum over-parameterization." NeurIPS2022.
> [7] Karhadkar, Kedar, Michael Murray, and Guido F. Montufar. "Bounds for the smallest eigenvalue of the NTK for arbitrary spherical data of arbitrary dimension." NeurIPS2024.
> [8] Liu, Chaoyue, Libin Zhu, and Mikhail Belkin. "Loss landscapes and optimization in over-parameterized non-linear systems and neural networks." _Applied and Computational Harmonic Analysis_ 59 (2022): 85-116.
> [9] Frei, Spencer, Niladri S. Chatterji, and Peter Bartlett. "Benign overfitting without linearity: Neural network classifiers trained by gradient descent for noisy linear data." COLT2022.
> [10] Han, X. Y., Vardan Papyan, and David L. Donoho. "Neural collapse under mse loss: Proximity to and dynamics on the central path." ICLR2022, Oral.
> [11] Zhou, Jinxin, et al. "On the optimization landscape of neural collapse under mse loss: Global optimality with unconstrained features." ICML2022.
> [12] Súkeník, Peter, Marco Mondelli, and Christoph H. Lampert. "Deep neural collapse is provably optimal for the deep unconstrained features model." NeurIPS2023.
> [13] Súkeník, Peter, Christoph Lampert, and Marco Mondelli. "Neural collapse vs. low-rank bias: Is deep neural collapse really optimal?." NeurIPS2024.

---

> ### Author Response · Authors · 2025-11-18
> **Reply to Reviewer ud5o (3/4)**
>
> ---
> > (Q4) I also think replacing the NC2 metric with a condition-number surrogate without justification is sloppy. The co-occurrence of NC1 and NC2 is critical for the normative statement of neural collapse to hold (theorem 3 in Papyan 2020). NC2 is strictly defined as equal angles and equal norms both satisfied. While when condition number = 1, the perfectly isotropic mean configuration coincides with ETF. Many non-ETF configurations can also yield small condition numbers.
>
> As we also emphasized in our response to Q1, it is important to note that **our paper does not aim to prove neural collapse**, nor do any of our theoretical results rely on NC2 or its surrogate. The NC2 (condition-number)  is used only as supplementary experimental evidence to illustrate the geometric structure of class means. It plays no role in our theoretical arguments.
>
> As you correctly pointed out, when the condition number equals $1$, meaning that class means have equal norms and are close to orthogonal, their configuration coincides with an ETF, after centering with the global mean exactly as in the NC2 definition of [Papyan+, 2020]. Thus, the condition-number surrogate we use is in fact a stronger requirement than the standard NC2 metric. This surrogate has also been employed in multiple neural collapse studies; see, for example, [1, 12, 13] from our Q3 references.
>
> To make this distinction clearer, **we have moved the definition of NC2 metric (previously in Section 4.2) into the experimental section. We have also revised the label in Figure 3 from "NC2 score" to "NC2 (condition-number)"** to avoid confusion between the original NC2 definition and our experimental surrogate. We are also open to moving the NC2 plots entirely to the appendix if you would prefer this additional step, and we are happy to make the presentation as clear as possible.
>
>
> ---
> > (Q5) Could flatness or implicit regularization, rather than NC1 dynamics, explain the delayed generalization you observe?
>
> This is an excellent question. In principle, delayed generalization could be explained by flatness or implicit reguralization if there exists a theory showing that their gradient-descent dynamics evolve on a time scale separate from the training-loss convergence, analogous to what Theorem 4.3 establishes for NC1 dynamics. Such an explanation would also require standing on prior work demonstrating that flatness or implicit reguralization improves generalization.
>
> Related to this point, as noted in our general comments, a recent paper that became public after our submission: [Flatness is Necessary, Neural Collapse is Not: Rethinking Generalization via Grokking, Han+, NeurIPS2025] may appear to contradict our claims at first glance. For completeness, and since you may find this relevant, We clarify here why this is not the case and highlight the novelty of our contribution.
> [Han+, 2025] also analyze grokking in relation to neural collapse, and additionally discuss flatness. Although the title suggests that neural collapse might not be relevant for generalization, this is **not** what actually argues. Rather, they state that neural collapse appears earlier than the acquisition of generalization. The emergence of neural collapse does not align with that of generalization, and therefore it does not fully explain generalization ability. In contrast, they observe that flatness correlates more closely with the decrease of test loss and thus appears to offer a better explanation in their experiments.
> However, this empirical observation is entirely consistent with our analysis. The apparent contradiction arises because [Han+, 2025] measure neural collapse using the *neural collapse clustering (NCC)* metric, which mixes multiple properties associated with neural collapse. In particular, NCC also captures the separation of class mean representations that naturally occurs through fitting the training set, and therefore it decreases earlier than the test loss. As we emphasize in Remark 4.2, our analysis isolates the contribution of within-class variance or RNC1 score, allowing us to separate out the cause of the generalization and correctly focus on the variance decrease that is actually responsible for the generalization emergence of grokking.
> Furthermore, the theoretical analysis in [Han+, 2025] is limited to showing that flatness is partially guaranteed under neural collapse. Their work does not address the connection to generalization nor analyze the associated training dynamics, and therefore **it does not provide an explanation for the delayed generalization phenomenon that you are specifically interested in**. In contrast, as illustrated in Figure 1, **our paper clarifies the relationships among several concepts, including information bottleneck, and it further characterizes their training dynamics through a neural-collapse-based analysis**. This leads to a unified understanding of the mechanisms underlying the grokking mechanism.

---

> ### Author Response · Authors · 2025-11-18
> **Reply to Reviewer ud5o (4/4)**
>
> ---
> > (Q6) Is Theorem 3.4 a tight bound or a loose ceiling?
>
> The short answer is that Theorem 3.4 itself is not a tight bound, but **its behavior closely tracks that of the tight upper bound**.
> The upper bound in Theorem 3.4 is derived from combining Lemma A.2 with the inequality $\log \det(A + I) \leq \mathrm{Tr}(A)$  for a positive definite matrix $A$. If we use only Lemma A.2, we obtain the following tight upper bound:
> $$
> I(Z;X \mid Y) \leq \frac{1}{2}\mathbb{E}\_{y\sim Y}\left[ \log\det \left( \frac{\mathrm{Cov}\_{X \mid Y=y}[\tilde{g}(X)]}{\sigma^2} + I \right) \right],
> $$
> which becomes exact when $Z \mid Y = y$ is Gaussian.
>
> Theorem 3.4 then replaces the log-determinant by its trace upper bound, which sufficiently reflects the behavior of the above bound.
> In particular, decreasing each coordinate of the rescaled within-class variance (in Theorem 3.4 bound) **always decreases the tight upper bound**. The potential concern with loose bounds is that making the bound small may not correspond to making the true quantity small, but it does not arise here.
>
> To clarify this formally, **we have added a new subsection, Appendix A.2.3: “Remark on Upper-bound Tightness”**, and **introduced a new proposition, Proposition A.3**, which provides a rigorous statement of the argument above in the revised version.

---

### Official Review · Reviewer_gZ1w · 2025-10-31

**Soundness:** 3
**Presentation:** 3
**Contribution:** 3
**Rating:** 6
**Confidence:** 2

**Summary:**

This paper demonstrates theoretically that the two training late-phase phenomena of grokking and information bottleneck (IB) dynamics are both due mainly to reduced population within-class variance.  It relates the measure of such variance reduction to the neural collapse degree present in the training dataset.  The paper shows that the model training duration ends sooner than the onset of the neural collapse progress, with the former dependent on the time taken to fit the training set and hence resulting in late-phase phenomena because neural collapse is responsible for such phenomena.  Experiments are conducted to validate its theoretical results.

**Strengths:**

This paper broadens the understanding of training dynamics by revealing the connection between late-phase phenomena and neural collapse.  It analyzes the variance of class-wise representations in the training data and shows that the decrease of empirical within-class variance leads to improved test accuracy, which takes place in the second compression phase to exhibit model grokking behavior after the first fitting phase.

By considering the grokking and IB dynamics as two representative late-phase phenomena of DNNs, the paper is the very first to demonstrate that both phenomena can be explained in terms of the population within-class variance, offering new insights into their underlying mechanisms.

The paper conducts a quantitative analysis on the discrepancy between the population within-class variance, allowing to evaluate the progression of neural collapse and implying a corresponding reduction in the population within-class variance to relate
the behaviors of grokking and IB dynamics to the development of neural collapse.

**Weaknesses:**

The experimental study for validating theoretical results is limited to DNN models with two classes, like the assumption of theoretical treatment.  It should be evaluated for the models targeting datasets with more classes, for generalization.

**Questions:**

Experiments on more general scenarios for models with more classes are desirable for result validation, since their results are not clear to hold.

---

> ### Author Response · Authors · 2025-11-18
> **Reply to Reviewer gZ1w**
>
> We truly appreciate your positive evaluation of our work. We hope that our response below will further increase your confidence in our work.
>
> ---
> > (Q1) The experimental study for validating theoretical results is limited to DNN models with two classes, like the assumption of theoretical treatment. It should be evaluated for the models targeting datasets with more classes, for generalization. Experiments on more general scenarios for models with more classes are desirable for result validation, since their results are not clear to hold.
>
> Our experiments supporting the theoretical results are conducted on MNIST and CIFAR10 (each with 10 classes) as well as on multiple text datasets (with 2, 4, and 6 classes). Therefore, the experiments are **not restricted to the two-class case**. In addition, **all theoretical results in this paper hold for a general number of classes $K$**, and are not limited to the binary case.
>
> Figure 2 focuses on the separating hyperplane of one pair of classes only to visualize margin evolution and within-class variance decrease in a multi-class classification problem. As you pointed out, this style of visualization may give the unintended impression that we conducted experiments for a two-class case. To avoid this misunderstanding, **we have added the following sentence at the end of the caption of Figure 2.** *"We additionally provide similar visualizations for several other class pairs in Appendix D.1."*
>
> **We have added the corresponding plots in Appendix D.1** in the revised version.

---

### Author Response · Authors · 2025-11-18

We have revised and uploaded the draft with modifications clearly colored for ease of reference.

In addition, after the submission of our paper, a related work [Flatness is Necessary, Neural Collapse is Not: Rethinking Generalization via Grokking, Han+, NeurIPS2025] was made public. Since the title may appear to contradict our claims at first glance, we clarify that their findings are fully consistent with ours and highlight the novelty of our contribution.

[Han+, 2025] discuss grokking in relation to neural collapse, which makes their setting similar to ours. Although the title suggests that neural collapse might not be relevant for generalization, this is **not** what it actually argues. Rather, they state that neural collapse appears earlier than the acquisition of generalization. The emergence of neural collapse does not align with that of generalization, and therefore it does not fully explain generalization ability. In contrast, they observe that flatness correlates more closely with the decrease in test loss and thus appears to offer a better explanation in their experiments.
However, **this empirical observation is entirely consistent with our analysis.** The apparent contradiction arises because [Han+, 2025] measure neural collapse using the *neural collapse clustering (NCC)* metric, which mixes multiple properties associated with neural collapse. In particular, NCC also captures the separation of class mean representations that naturally occurs through fitting the training set, and therefore it decreases earlier than the test loss. As we emphasize in Remark 4.2, our analysis isolates the contribution of within-class variance or RNC1 score, allowing us to separate out the cause of the generalization and correctly focus on the variance decrease that is actually responsible for the emergence of generalization in grokking.
Furthermore, the theoretical analysis in [Han+, 2025] is limited to showing that flatness is partially guaranteed under neural collapse. Their work address neither the connection to generalization nor the associated training dynamics. In contrast, as illustrated in Figure 1, our paper clarifies the relationships among several concepts, including information bottleneck, and it further characterizes their training dynamics through a neural-collapse-based analysis. This leads to a unified understanding of the mechanisms underlying grokking.

---

### Meta-Review · Area_Chair_CnTh · 2025-12-31

**Summary:**

This paper studies the phenomena of grokking and information bottleneck (IB) through the lens of neural collapse: the contraction of the within-class variance in representations is identified as the culprit of both grokking and IB. For grokking, the authors derive an upper bound on generalization error based on the population within-class variance. For IB, the authors show that the redundant information discarded in the IB compression phase is also bounded by the population within-class variance. Next, it is shown that the empirical within-class variance concentrates to the population one, and the former is then linked to neural collapse. The link can be made formal by tracking the dynamics of gradient descent, following the approach by [Jacot et al., 2025].

The three reviewers appreciated the novel connection between grokking, IB and neural collapse. A number of weaknesses have been raised as well: the heuristic nature of some claims, the strong assumptions, the role of NC2, the behavior under different normalization, the connection to related works. Based on my reading of the exchanges, my opinion is that most of these issues have been resolved. Another issue -- not raised by the reviewers, but coming from my own reading of the manuscript -- lies in the technical novelty: the part on the gradient descent dynamics appears incremental with respect to [Jacot et al., 2025]. Having said that, the connection between neural collapse and grokking+IB is, at the best of my knowledge, new, and it will be of interest to the ICLR community. Thus, I am happy to recommend acceptance.

**Reviewer Concerns:**

Reviewer gZ1w was overall positive and the only concern about multiple classes was addressed properly.

Reviewer ud5o raised several points, which I believe were properly addressed by the authors. Some are of subjective nature, eg the strong assumptions, but in my opinion this weakness does not justify a rejection.

Reviewer oBWN raised several objections as well, but was satisfied by the response of the authors. I agree that most of the points raised have been addressed in a satisfactory manner (an exception is, in my opinion, Q1 about different kinds of normalization, but one can indeed argue that this is out of scope of the present contribution).

**Reviewer Scores:**

One of the reviewers had already raised the score and I find it possible (if not likely) that a consensus could have been reached towards accepting the paper. After my own reading of the manuscript, this is in fact my stand.

---

### Decision · Program_Chairs · 2026-01-26

Accept (Poster)